# Graph-constrained Reasoning: Faithful Reasoning on Knowledge Graphs with Large Language Models

## Abstract

Large language models (LLMs) have demonstrated impressive reasoning abilities, but they still struggle with faithful reasoning due to knowledge gaps and hallucinations. To address these issues, knowledge graphs (KGs) have been utilized to enhance LLM reasoning through their structured knowledge. However, existing KG-enhanced methods, either retrieval-based or agent-based, encounter difficulties in accurately retrieving knowledge and efficiently traversing KGs at scale. In this work, we introduce graph-constrained reasoning (GCR), a novel framework that bridges structured knowledge in KGs with unstructured reasoning in LLMs. To eliminate hallucinations, GCR ensures faithful KG-grounded reasoning by integrating KG structure into the LLM decoding process through KG-Trie, a trie-based index that encodes KG reasoning paths. KG-Trie constrains the decoding process, allowing LLMs to directly reason on graphs and generate faithful reasoning paths grounded in KGs. Additionally, GCR leverages a lightweight KG-specialized LLM for graph-constrained reasoning alongside a powerful general LLM for inductive reasoning over multiple reasoning paths, resulting in accurate reasoning with zero reasoning hallucination. Extensive experiments on several KGQA benchmarks demonstrate that GCR achieves state-of-the-art performance and exhibits strong zero-shot generalizability to unseen KGs without additional training.

## 1 Introduction

Large language models (LLMs) have shown impressive reasoning abilities in handling complex tasks (Qiao et al., 2023; Huang & Chang, 2023), marking a significant leap that bridges the gap between human and machine intelligence. However, LLMs still struggle with conducting faithful reasoning due to issues of *lack of knowledge* and *hallucination* (Huang et al., 2024; Wang et al., 2023). These issues result in factual errors and flawed reasoning processes (Nguyen et al., 2024), which greatly undermine the reliability of LLMs in real-world applications.

To address these issues, many studies utilize knowledge graphs (KGs), which encapsulate extensive factual information in a structured format, to improve the reasoning abilities of LLMs (Pan et al., 2024; Luo et al., 2024). Nevertheless, because of the unstructured nature of LLMs, directly applying them to reason on KGs is challenging.

Existing KG-enhanced LLM reasoning methods can be roughly categorized into two groups: *retrieval-based* and *agent-based* paradigms, as shown in Figure 2 (a) and (b). Retrieval-based methods (Li et al., 2023; Yang et al., 2024b; Dehghan et al., 2024) retrieve relevant facts from KGs with an external retriever and then feed them into the inputs of LLMs for reasoning. Agent-based methods (Sun et al., 2024; Zhu et al., 2024; Jiang et al., 2024) treat LLMs as agents that iteratively interact with KGs to find reasoning paths and answers.

Despite their success, retrieval-based methods require additional accurate retrievers, which may not generalize well to unseen questions or account for the graph structure (Mavromatis & Karypis, 2024). Conversely, agent-

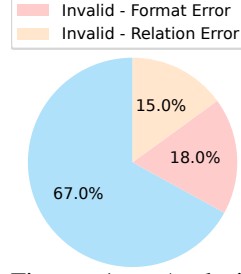

Figure 1: Analysis of reasoning errors in RoG (Luo et al., 2024).

(legend) Faithful Reasoning Path; Invalid - Format Error; Invalid - Relation Error
15.0%; 18.0%; 67.0%

based methods necessitate multiple rounds of interaction between agents and KGs, leading to high computational costs and latency (Dehghan et al., 2024). Furthermore, existing works still suffer from serious hallucination issues (Agrawal et al., 2024). Sui et al. (2024) indicates that RoG (Luo et al., 2024), a leading KG-enhanced reasoning method, still experiences 33% hallucination errors during reasoning on KGs, as shown in Figure 1.

To this end, we introduce graph-constrained reasoning (GCR), a novel KG-guided reasoning paradigm that connects unstructured reasoning in LLMs with structured knowledge in KGs, seeking to eliminate hallucinations during reasoning on KGs and ensure faithful reasoning. Inspired by the concept that LLMs reason through decoding (Wei et al., 2022), we incorporate the KG structure into the LLM decoding process. This enables LLMs to directly reason on graphs by generating reliable reasoning paths grounded in KGs that lead to correct answers.

In GCR, we first convert KG into a structured index, KG-Trie, to facilitate efficient reasoning on KG using LLM. Trie is also known as the prefix tree (Wikipedia contributors, 2024) that compresses a set of strings, which can be used to restrict LLM output tokens to those starting with valid prefixes (De Cao et al., 2022; Xie et al., 2022). KG-Trie encodes the reasoning paths in KGs as formatted strings to constrain the decoding process of LLMs. Then, we propose graph-constrained decoding that employs a lightweight KG-specialized LLM to generate multiple KG-grounded reasoning paths and hypothesis answers. With the constraints from KG-Trie, we ensure faithful reasoning while leveraging the strong reasoning capabilities of LLMs to efficiently explore paths on KGs in constant time. Finally, we input multiple generated reasoning paths and hypothesis answers into a powerful general LLM to utilize its inductive reasoning ability to produce final answers. In this way, GCR combines the graph reasoning strength of KG-specialized LLMs and the inductive reasoning advantage in general LLMs to achieve faithful and accurate reasoning on KGs. The main contributions of this work are as follows:

- We propose a novel framework called graph-constrained reasoning (GCR) that bridges the gap between structured knowledge in KGs and unstructured reasoning in LLMs, allowing for efficient reasoning on KGs via LLM decoding.

- We combine the complementary strengths of a lightweight KG-specialized LLM with a powerful general LLM to enhance reasoning performance by leveraging their respective graph-based reasoning and inductive reasoning capabilities.

- We conduct extensive experiments on several KGQA reasoning benchmarks, demonstrating that GCR not only achieves state-of-the-art performance with zero hallucination, but also shows zero-shot generalizability for reasoning on unseen KGs without additional training.

## 2  RELATED WORK

**LLM reasoning.** Many studies have been proposed to analyze and improve the reasoning ability of LLMs (Wei et al., 2022; Wang et al., 2024; Yao et al., 2024). To elicit the reasoning ability of LLMs, Chain-of-thought (CoT) reasoning (Wei et al., 2022) prompts the model to generate a chain of reasoning steps in response to a question. Wang et al. (2024) propose a self-consistency mechanism that generates multiple reasoning paths and selects the most consistent answer across them. The tree-of-thought (Yao et al., 2024) structures reasoning as a branching process, exploring multiple steps in a tree-like structure to find optimal solutions. Other studies focus on fine-tuning LLMs on various reasoning tasks to improve reasoning abilities (Yu et al., 2022; Hoffman et al., 2024). For instance, OpenAI (2024c) adopts reinforcement learning to train their most advanced LLMs called "OpenAI o1" to perform complex reasoning, which produces a long internal chain of thought before final answers.

**KG-enhanced LLM reasoning.** To mitigate the knowledge gap and hallucination issues in LLM reasoning, research incorporates KGs to enhance LLM reasoning (Pan et al., 2024). KD-CoT (Wang et al., 2023) retrieve facts from an external knowledge graph to guide the CoT performed by LLMs. RoG (Luo et al., 2024) proposes a planning-retrieval-reasoning framework that retrieves reasoning paths from KGs to guide LLMs conducting faithful reasoning. To capture graph structure, GNN-RAG (Mavromatis & Karypis, 2024) adopts a lightweight graph neural network to effectively retrieve from KGs. Instead of retrieving, StructGPT (Jiang et al., 2023) and ToG (Sun et al., 2024) treat LLMs as agents to interact with KGs to find reasoning paths leading to the correct answers.

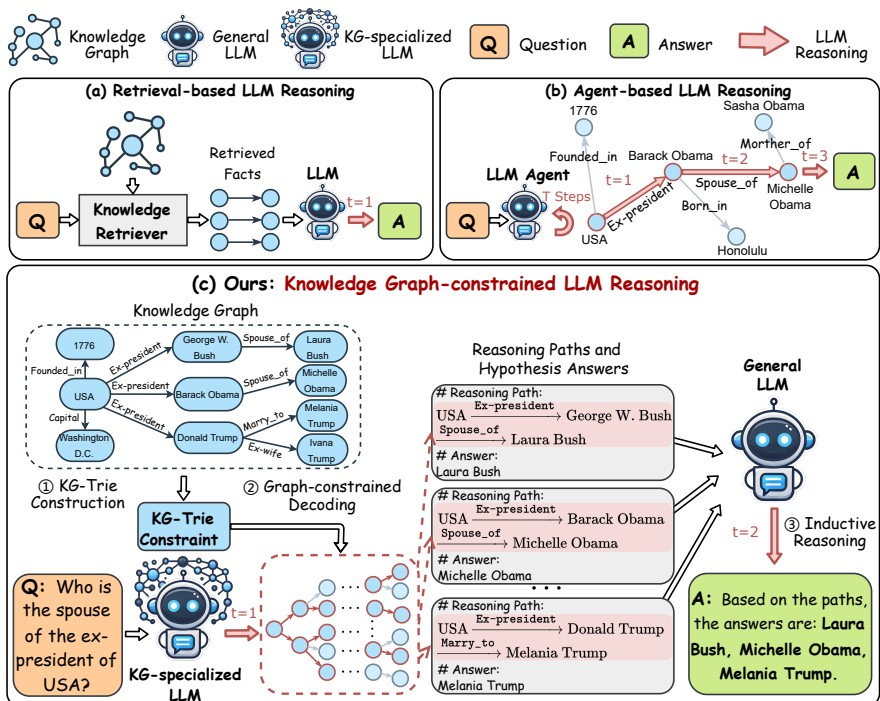

Figure 2: Illustration of existing KG-enhanced LLM reasoning paradigms and proposed graph-constrained reasoning (GCR). 1) First, given a KG, we convert it into the KG-Trie, serving as a structured index to facilitate efficient reasoning path searches using LLMs. 2) Then, we design a graph-constrained decoding process that employs a lightweight KG-specialized LLM to generate multiple KG-grounded reasoning paths and hypothesis answers. This ensures the faithfulness of the reasoning process while leveraging the strong capabilities of LLMs to efficiently explore reasoning paths within KGs. 3) Finally, we input the generated reasoning paths and hypothesis answers into a powerful general LLM to utilize its inductive reasoning ability to produce final answers.

## 3 PRELIMINARY

**Knowledge Graphs (KGs)** represent a wealth of factual knowledge as a collection of triples: $\mathcal{G} = \{(e, r, e') \in \mathcal{E} \times \mathcal{R} \times \mathcal{E}\}$, where $\mathcal{E}$ and $\mathcal{R}$ denote the set of entities and relations, respectively.

**Reasoning Paths** are sequences of consecutive triples in KGs: $\boldsymbol{w_z} = e_0 \xrightarrow{r_1} e_1 \xrightarrow{r_2} \ldots \xrightarrow{r_l} e_l$, where $\forall (e_{i-1}, r_i, e_i) \in \mathcal{G}$. The paths reveal the connections between knowledge that potentially facilitate reasoning. For example, the reasoning path: $\boldsymbol{w_z} = \text{Alice} \xrightarrow{\texttt{marry\_to}} \text{Bob} \xrightarrow{\texttt{father\_of}}$ Charlie indicates that "Alice" is married to "Bob" and "Bob" is the father of "Charlie". Therefore, "Alice" could be reasoned to be the mother of "Charlie".

**Knowledge Graph Question Answering (KGQA)** is a representative reasoning task with the assistance of KGs. Given a natural language question $q$ and a KG $\mathcal{G}$, the task aims to design a function $f$ to reason answers $a \in \mathcal{A}$ based on knowledge from $\mathcal{G}$, i.e., $a = f(q, \mathcal{G})$.

**KG-constrained Zero-hallucination.** As facts in KGs are usually verified, making them a reliable source for assessing the faithfulness of LLM reasoning (Nguyen et al., 2024). In this paper, we define KG-constrained zero hallucinations as the LLM generated reasoning paths can be fully grounded within KGs, ensuring the alignment of reasoning process with real-world facts. The limitations of the definition are discussed in Appendix G.

## 4 APPROACH

### 4.1 FROM CHAIN-OF-THOUGHT REASONING TO GRAPH-CONSTRAINED REASONING

**Chain-of-Thought Reasoning (CoT)** (Wei et al., 2022) has been widely adopted to enhance the reasoning ability of LLMs by autoregressively generating a series of reasoning steps leading to the answer. Specifically, given a question $q$, CoT models the joint probability of the answer $a$ and

reasoning steps $z$ as

$$P(a|q) = \sum_{\boldsymbol{z}} P_\theta(a|\boldsymbol{z}, q) P_\theta(\boldsymbol{z}|q) = \sum_{\boldsymbol{z}} P_\theta(a|q, \boldsymbol{z}) \prod_{i=1}^{|\boldsymbol{z}|} P_\theta(z_i|q, z_{1:i-1}), \tag{1}$$

where $q$ denotes the input question, $a$ denotes the final answer, $\theta$ denotes the parameters of LLMs, and $z_i$ denotes the $i$-th step of the reasoning process $\boldsymbol{z}$. To further enhance the reasoning ability, many previous works focus on improving the reasoning process $P_\theta(\boldsymbol{z}|q)$ by exploring and aggregating multiple reasoning processes (Wang et al., 2024; Yao et al., 2024).

Despite the effectiveness, a major issue remains the faithfulness of the reasoning process generated by LLMs (Huang et al., 2024). The reasoning is represented as a sequence of tokens decoded step-by-step, which can accumulate errors and result in hallucinated reasoning paths and answers (Nguyen et al., 2024). To address these issues, we utilize knowledge graphs (KGs) to guide LLMs toward faithful reasoning.

**KG-enhanced Reasoning** utilizes the structured knowledge in KGs to improve the reasoning of LLMs (Luo et al., 2024; Sun et al., 2024), which can generally be expressed as finding a reasoning path $\boldsymbol{w_z}$ on KGs that connects the entities mentioned in the question and the answer. This can be formulated as

$$P(a|q, \mathcal{G}) = \sum_{\boldsymbol{w_z}} P_\phi(a|q, \boldsymbol{w_z}) P_\phi(\boldsymbol{w_z}|q, \mathcal{G}), \tag{2}$$

where $P_\phi(\boldsymbol{w_z}|q, \mathcal{G})$ denotes the probability of discovering a reasoning path $\boldsymbol{w_z}$ on KGs $\mathcal{G}$ given the question $q$ by a function parameterized by $\phi$. To acquire reasoning paths for reasoning, most prior studies follow the retrieval-based (Li et al., 2023) or agent-based paradigm (Sun et al., 2024), as shown in Figure 2 (a) and (b), respectively. Nevertheless, retrieval-based methods rely on precise additional retrievers, while agent-based methods are computationally intensive and lead to high latency. To address these issues, we propose a novel graph-constrained reasoning paradigm (GCR).

**Graph-constrained Reasoning (GCR)** directly incorporates KGs into the decoding process of LLMs to achieve faithful reasoning. The overall framework of GCR is illustrated in Figure 2 (c), which consists of three main components: 1) Knowledge Graph Trie Construction: building a structural index of KG to guide LLM reasoning, 2) Graph-constrained Decoding: generating KG-grounded paths and hypothesis answers using LLMs, and 3) Graph Inductive Reasoning: reasoning over multiple paths and hypotheses to derive final answers.

## 4.2 KNOWLEDGE GRAPH TRIE CONSTRUCTION

Knowledge graphs (KGs) store abundant knowledge in a structured format. However, large language models (LLMs) struggle to efficiently access and reason on KGs due to their unstructured nature. To address this issue, we propose to convert KGs into knowledge graph Tries (KG-Tries), which serve as a structured index of KGs to facilitate efficient reasoning on graphs using LLMs.

A Trie (a.k.a. prefix tree) (Wikipedia contributors, 2024; Fredkin, 1960) is a tree-like data structure that stores a dynamic set of strings, where each node represents a common prefix of its children. Tries can be used to restrict LLM output tokens to those starting with valid prefixes (De Cao et al., 2022; Xie et al., 2022; Chen et al., 2022). The tree structure of Trie is an ideal choice for encoding the reasoning paths in KGs for LLMs to efficiently traverse.

Given a KG $\mathcal{G}$ and a question $q$, we first retrieve paths $\mathcal{W_z}$ within $L$ hops starting from entities mentioned in the question $\{e_q\}$. We adopt the breadth-first search (BFS) algorithm to retrieve reasoning paths, but it can be replaced with other efficient graph-traversing algorithms, such as random walk (Xia et al., 2019). The retrieved paths are formatted as sentences using the template shown in Figure 8. The formatted sentences are then split into tokens by the tokenizer of LLM and stored as a KG-Trie $\mathcal{C_G}$. The overall process can be formulated as:

$$\mathcal{W_z} = \text{BFS}(\mathcal{G}, \{e_q\}, L), \tag{3}$$

$$\mathcal{T_z} = \text{Tokenizer}(\mathcal{W_z}), \tag{4}$$

$$\mathcal{C_G} = \text{Trie}(\mathcal{T_z}), \tag{5}$$

where $e_q$ denotes the entities mentioned in the question, $L$ denotes the maximum hops of paths, and $\mathcal{T_z}$ denotes the tokens of reasoning paths. The KG-Trie $\mathcal{C_G}$ is used as a constraint to guide the LLM decoding process.

By constructing KG-Trie for each question entity, we can enable efficient traversal of reasoning paths in constant time ($O(|\mathcal{W}_{\boldsymbol{z}}|)$) without costly graph traversal (Sun et al., 2024). Moreover, KG-Trie can be pre-constructed offline and loaded during reasoning for fast inference, or it can be built on-demand to reduce pre-processing time. Detailed discussions on construction efficiency and potential solutions for further improvements to scale into real-world applications is available in Appendix B. This significantly reduces the computational cost and latency of reasoning on KGs, making it feasible for real-time applications.

---

=========================== Prompt Input ===============================
Please generate some reasoning paths in the KG starting from the topic entities to answer the question.
# Question: what is the name of justin bieber brother?

=========================== LLM Output =================================
# Reasoning Path: `<PATH>` Justin Bieber → people.person.parents → Jeremy Bieber → people.person.children → Jaxon Bieber `</PATH>`
# Answer: Jaxon Bieber

---

Figure 3: An example of the graph-constrained decoding. Detailed prompts can be found in Figure 9.

## 4.3 GRAPH-CONSTRAINED DECODING

Large language models (LLMs) have strong reasoning capabilities but still suffer from severe hallucination issues, which undermines the trustworthiness of the reasoning process. To tackle this issue, we propose graph-constrained decoding, which unifies the reasoning ability of LLMs with the structured knowledge in KGs to generate faithful KG-grounded reasoning paths leading to answers.

Given a question $q$, we design an instruction prompt to harness the reasoning ability of LLMs to generate reasoning paths $\boldsymbol{w}_{\boldsymbol{z}}$ and hypothesis answers $a$. To eliminate the hallucination during reasoning on KGs, we adopt the KG-Trie $\mathcal{C}_{\mathcal{G}}$ as constraints to guide the decoding process of LLMs and only generate reasoning paths that are valid in KGs, formulated as:

$$P_\phi(a, \boldsymbol{w}_{\boldsymbol{z}}|q) = \underbrace{P_\phi(a|q, \boldsymbol{w}_{\boldsymbol{z}})}_{\text{Regular decoding}} \overbrace{\prod_{i=1}^{|\boldsymbol{w}_{\boldsymbol{z}}|} P_\phi(w_{z_i}|q, w_{z_{1:i-1}}) \mathcal{C}_{\mathcal{G}}(w_{z_i}|w_{z_{1:i-1}})}^{\text{Graph-constrained decoding}}, \quad (6)$$

$$\mathcal{C}_{\mathcal{G}}(w_{z_i}|w_{z_{1:i-1}}) = \begin{cases} 1, \exists \text{prefix}(w_{z_{1:i}}, \boldsymbol{w}_{\boldsymbol{z}}), \exists \boldsymbol{w}_{\boldsymbol{z}} \in \mathcal{W}_{\boldsymbol{z}}, \\ 0, else, \end{cases} \quad (7)$$

where $w_{z_i}$ denotes the $i$-th token of the reasoning path $\boldsymbol{w}_{\boldsymbol{z}}$, $P_\phi$ denotes the token probabilities predicted by the LLM with parameters $\phi$, and $\mathcal{C}_{\mathcal{G}}(w_{z_i}|w_{z_{1:i-1}})$ denotes the constraint function that checks whether the generated tokens $w_{z_{1:i}}$ is a valid prefix of the reasoning path using KG-Trie. After a valid reasoning path is generated, we switch back to the regular decoding process to generate a hypothesis answer conditioned on the path.

To further enhance KG reasoning ability, we fine-tune a lightweight KG-specialized LLM with parameters $\phi$ on the graph-constrained decoding task. Specifically, given a question $q$, the LLM is optimized to generate relevant reasoning paths $\boldsymbol{w}_{\boldsymbol{z}}$ that are helpful for answering the question, then provide a hypothesis answer $a$ based on it, which can be formulated as:

$$\mathcal{L} = \mathbb{E}_{(q, \boldsymbol{w}_{\boldsymbol{z}}, a) \sim \mathcal{D}_{\mathcal{G}}} \log P_\phi(a, \boldsymbol{w}_{\boldsymbol{z}}|q) = \mathbb{E}\left[ \log \prod_{i=1}^{|a|} P_\phi(a_i|q, \boldsymbol{w}_{\boldsymbol{z}}, a_{1:i-1}) \prod_{j=1}^{|\boldsymbol{w}_{\boldsymbol{z}}|} P_\phi(w_{z_j}|q, w_{z_{1:j-1}}) \right],$$
$$(8)$$

where $a_i$ and $w_{z_j}$ denote the $i$-th token of the answer $a$ and the $j$-th token of the reasoning path $\boldsymbol{w}_{\boldsymbol{z}}$, respectively.

The training data $(q, \boldsymbol{w}_{\boldsymbol{z}}, a) \in \mathcal{D}_{\mathcal{G}}$ consists of question-answer pairs and reasoning paths generated from KGs. We use the shortest paths connecting the entities in the question and answer as the reasoning path $\boldsymbol{w}_{\boldsymbol{z}}$ for training, where details can be found in Appendix C. An example of graph-constrained decoding is illustrated in Figure 3, where `<PATH>` and `</PATH>` are special tokens to control the start and end of graph-constrained decoding. Experiment results in Section 5.2 show that even a lightweight KG-specialized LLM (0.5B) can achieve satisfactory performance in KG reasoning.

The graph-constrained decoding method differs from retrieval-based methods by integrating a pre-constructed KG-Trie into the decoding process of LLMs. This not only reduces input tokens, but also bridges the gap between unstructured reasoning in LLMs and structured knowledge in KGs, allowing for efficient reasoning on KGs regardless of its scale, which results in faithful reasoning leading to answers. Additionally, experimental results in Section 5.4 demonstrate that KG-Trie can integrate with new KGs on the fly, showcasing its zero-shot generalizability for reasoning on unseen KGs without further training.

### 4.4 GRAPH INDUCTIVE REASONING

Graph-constrained decoding harnesses the reasoning ability of a KG-specialized LLM to generate a faithful reasoning path and a hypothesis answer. However, complex reasoning tasks typically admit multiple reasoning paths that lead to correct answers (Stanovich et al., 2000). Incorporating diverse reasoning paths would be beneficial for deliberate thinking and reasoning (Evans, 2010; Wang et al., 2024). To this end, we propose to input multiple reasoning paths and hypothesis answers generated by the KG-specialized LLM into a powerful general LLM to leverage its inductive reasoning ability to produce final answers.

The graph-constrained decoding seamlessly integrates into the decoding process of LLMs, allowing it to be paired with various LLM generation strategies like beam-search (Federico et al., 1995) to take advantage of the GPU parallel computation. Thus, given a question, we adopt graph-constrained decoding to simultaneously generate $K$ reasoning paths and hypothesis answers with beam search in a single LLM call, which are then inputted into a general LLM to derive final answers. The overall process can be formulated as:

$$\mathcal{Z}_K = \{a^k, \boldsymbol{w}_{\boldsymbol{z}}^k\}_{k=1}^K = \arg \text{top-}K \, P_\phi(a, \boldsymbol{w}_{\boldsymbol{z}}|q), \tag{9}$$

$$P_\theta(\mathcal{A}|q, \mathcal{Z}_K) \simeq \prod_{k=1}^K P_\theta(\mathcal{A}|q, a^k, \boldsymbol{w}_{\boldsymbol{z}}^k), \tag{10}$$

where $\theta$ denotes the parameters of the general LLM, $\mathcal{Z}_K$ denotes the set of top-$K$ reasoning paths and hypothesis answers, and $\mathcal{A}$ denotes the final answers.

We follow the FiD framework (Izacard & Grave, 2021; Singh et al., 2021) to incorporate multiple reasoning paths and hypothesis answers to conduct inductive reasoning within one LLM call, i.e., $P_\theta(\mathcal{A}|q, \mathcal{Z}_K)$, where detailed prompts can be found in Figure 10. The general LLM can be any powerful LLM, such as ChatGPT (OpenAI, 2022), or Llama-3 (Meta, 2024), which can effectively leverage their internal reasoning ability to reason over multiple reasoning paths to produce final answers without additional fine-tuning.

## 5 EXPERIMENT

In our experiments, we aim to answer the following research questions: **RQ1:** Can GCR achieve state-of-the-art reasoning performance with balances between efficiency and effectiveness? **RQ2:** Can GCR eliminate hallucinations and conduct faithful reasoning? **RQ3:** Can GCR generalize to unseen KGs on the fly?

### 5.1 EXPERIMENT SETUPS

**Datasets.** Following previous research (Luo et al., 2024; Sun et al., 2024), we first evaluate the reasoning ability of GCR on two benchmark KGQA datasets: WebQuestionSP (WebQSP) (Yih et al., 2016) and Complex WebQuestions (CWQ) (Talmor & Berant, 2018). Freebase (Bollacker et al., 2008) is adopted as the knowledge graph for both datasets. To further evaluate the generalizability of GCR, we conduct zero-shot transfer experiments on three new KGQA datasets: FreebaseQA (Jiang et al., 2019), CSQA (Talmor et al., 2019) and MedQA (Jin et al., 2021). FreebaseQA adopts the same Freebase KG. For CSQA, we use ConceptNet (Speer et al., 2017) as the KG, while for MedQA, we use a medical KG constructed from the Unified Medical Language System (Yasunaga et al., 2021). The details of the datasets are described in Appendix C.

**Baselines.** We compare GCR with the 22 baselines grouped into three categories: 1) *LLM reasoning methods*, 2) *graph reasoning methods*, and 3) *KG-enhanced LLM reasoning methods*. The detailed baselines are listed in Appendix D.

Table 1: Performance comparison with different baselines on the two KGQA datasets.

| Types | Methods | WebQSP | | CWQ | |
|---|---|---|---|---|---|
| | | Hit | F1 | Hit | F1 |
| LLM Reasoning | Qwen2-0.5B (Yang et al., 2024a) | 26.2 | 17.2 | 12.5 | 11.0 |
| | Qwen2-1.5B (Yang et al., 2024a) | 41.3 | 28.0 | 18.5 | 15.7 |
| | Qwen2-7B (Yang et al., 2024a) | 50.8 | 35.5 | 25.3 | 21.6 |
| | Llama-2-7B (Touvron et al., 2023) | 56.4 | 36.5 | 28.4 | 21.4 |
| | Llama-3.1-8B (Meta, 2024) | 55.5 | 34.8 | 28.1 | 22.4 |
| | GPT-4o-mini (OpenAI, 2024a) | 63.8 | 40.5 | 63.8 | 40.5 |
| | ChatGPT (OpenAI, 2022) | 59.3 | 43.5 | 34.7 | 30.2 |
| | ChatGPT+Few-shot (Brown et al., 2020) | 68.5 | 38.1 | 38.5 | 28.0 |
| | ChatGPT+CoT (Wei et al., 2022) | 73.5 | 38.5 | 47.5 | 31.0 |
| | ChatGPT+Self-Consistency (Wang et al., 2024) | 83.5 | 63.4 | 56.0 | 48.1 |
| Graph Reasoning | GraftNet (Sun et al., 2018) | 66.7 | 62.4 | 36.8 | 32.7 |
| | NSM (He et al., 2021) | 68.7 | 62.8 | 47.6 | 42.4 |
| | SR+NSM (Zhang et al., 2022) | 68.9 | 64.1 | 50.2 | 47.1 |
| | ReaRev (Mavromatis & Karypis, 2022) | 76.4 | 70.9 | 52.9 | 47.8 |
| | UniKGQA (Jiang et al., 2022) | 77.2 | 72.2 | 51.2 | 49.1 |
| KG+LLM | KD-CoT (Wang et al., 2023) | 68.6 | 52.5 | 55.7 | - |
| | EWEK-QA (Dehghan et al., 2024) | 71.3 | - | 52.5 | - |
| | ToG (ChatGPT) (Sun et al., 2024) | 76.2 | - | 57.6 | - |
| | ToG (GPT-4) (Sun et al., 2024) | 82.6 | - | 68.5 | - |
| | EffiQA (Dong et al., 2024) | 82.9 | - | 69.5 | |
| | RoG (Llama-2-7B) (Luo et al., 2024) | 85.7 | 70.8 | 62.6 | 56.2 |
| | GNN-RAG (Mavromatis & Karypis, 2024) | 85.7 | 71.3 | 66.8 | 59.4 |
| | GNN-RAG+RA (Mavromatis & Karypis, 2024) | 90.7 | 73.5 | 68.7 | 60.4 |
| | GCR (Llama-3.1-8B + ChatGPT) | **92.6** | 73.2 | 72.7 | 60.9 |
| | GCR (Llama-3.1-8B + GPT-4o-mini) | 92.2 | **74.1** | **75.8** | **61.7** |

**Evaluation Metrics.** We adopt Hit and F1 as the evaluation metrics following previous works (Luo et al., 2024; Sun et al., 2024) on WebQSP and CWQ. Hit checks whether any correct answer exists in the generated predictions, while F1 considers the coverage of all answers by balancing the precision and recall of predictions. Because CSQA and MedQA are multiple-choice QA datasets, we adopt accuracy as the evaluation metric.

**Implementations.** For GCR, we use the KG-Trie to index all the reasoning paths within 2 hops starting from question entities. For the LLMs, we use a fine-tuned Llama-3-8B (Meta, 2024) as the KG-specialized LLM. We generate top-10 reasoning paths and hypothesis answers from graph-constrained decoding. We adopt the advanced ChatGPT (OpenAI, 2022) and GPT-4o-mini (OpenAI, 2024a) as the general LLMs for inductive reasoning. The detailed hyperparameters and experiment settings are described in Appendix E.

## 5.2 RQ1: REASONING PERFORMANCE AND EFFICIENCY

**Main Results.** In this section, we compare GCR with other baselines on KGQA benchmarks to evaluate the reasoning performance. From the results shown in Table 1, GCR achieves the best performance on both datasets, outperforming the second-best by 2.1% and 9.1% in terms of Hit on WebQSP and CWQ, respectively. The results demonstrate that GCR can effectively leverage KGs to enhance LLMs and achieve state-of-the-art reasoning performance.

Among the LLM reasoning methods, ChatGPT with self-consistency prompts demonstrates the best performance, which indicates the powerful reasoning ability inherent in LLMs. However, their performances are still limited by the model size and complex reasoning required over structured data. Graph reasoning methods, such as ReaRev, achieve competitive performance on WebQSP by explicitly modeling the graph structure. But they struggle to generalize across different datasets and underperform on CWQ. In KG+LLM methods, both agent-based methods (e.g., ToG, EffiQA) and retrieval-based methods (e.g., RoG, GNN-RAG) achieve the second-best performance. Nevertheless, they still suffer from inefficiency and reasoning hallucinations which limit their performance. In contrast, GCR effectively eliminates hallucinations and conducts faithful reasoning by leveraging the structured KG index and graph-constrained decoding.

**Efficiency Analysis.** To show the efficiency of GCR, we compare the average runtime, number of LLM calls, and number of input tokens with retrieval-based and agent-based methods in Table 2. For retrieval-based methods, we compare with dense retrievers (e.g., S-Bert (Reimers & Gurevych, 2019), BGE (Zhang et al., 2023), OpenAI-Emb. (OpenAI, 2024b)) and graph-based retrievers (e.g.,

Table 2: Efficiency and performance comparison of different methods on WebQSP.

| Types | Methods | Hit | Avg. Runtime (s) | Avg. # LLM Calls | Avg. # LLM Tokens |
|---|---|---|---|---|---|
| Retrieval-based | S-Bert | 66.9 | 0.87 | 1 | 293 |
| | BGE | 72.7 | 1.05 | 1 | 357 |
| | OpenAI-Emb. | 79.0 | 1.77 | 1 | 330 |
| | GNN-RAG | 85.7 | 1.52 | 1 | 414 |
| | RoG | 85.7 | 2.60 | 2 | 521 |
| Agent-based | ToG | 75.1 | 16.14 | 11.6 | 7,069 |
| | EffiQA | 82.9 | - | 7.3 | - |
| Ours | GCR | **92.6** | 3.60 | 2 | 231 |

GNN-RAG (Mavromatis & Karypis, 2024), RoG (Luo et al., 2024)), which retrieve reasoning paths from KGs and feed them into LLMs for reasoning answers. For agent-based methods, we compare with ToG (Sun et al., 2024) and EffiQA[1] (Dong et al., 2024), which heuristically search on KGs for answers. The detailed settings are described in Appendix E.

Dense retrievers are most efficient in terms of runtime and LLM calls as they convert all paths into sentences and encode them as embeddings in advance. However, they sacrifice their accuracy in retrieving as they are not designed to encode graph structure. Graph-based retrievers and agent-based methods achieve better performance by considering graph structure; however, they require more time and LLM calls. Specifically, the retrieved graph is fed as inputs to LLMs, which leads to a large number of input tokens. Agent-based methods, like ToG, require more LLM calls and input tokens as the question difficulty increases due to their iterative reasoning process. In contrast, GCR achieves the best performance with a reasonable runtime and number of LLM calls. With the help of KG-Trie, GCR explores multiple reasoning paths at the same time during the graph-constrained decoding, which does not involve additional LLM calls or input tokens and benefits from the parallel GPU computation with low latency. More efficiency analysis under different beam sizes used for graph-constrained decoding can be found in parameter analysis.

**Ablation Study.** We first conduct an ablation study to analyze the effectiveness of the KG-specialized LLM and general LLM in GCR. As shown in Table 3, the full GCR achieves the best performance on both datasets.

Table 3: Ablation studies of GCR on two KGQA datasets.

| Variants | WebQSP | | | CWQ | | |
|---|---|---|---|---|---|---|
| | F1 | Precision | Recall | F1 | Precision | Recall |
| GCR (Llama-3.1-8B + ChatGPT) | **73.2** | **80.0** | **76.9** | **60.9** | **61.1** | **66.6** |
| GCR w/o KG-specialized LLM | 52.9 | 66.3 | 50.2 | 37.5 | 40.8 | 37.9 |
| GCR w/o General LLM | 57.0 | 58.0 | 70.1 | 39.4 | 32.8 | 64.3 |

By removing the KG-specialized LLM, we feed all 2-hop reasoning paths into the general LLM. This results in a significant performance drop, indicating its importance in utilizing reasoning ability to find relevant paths on KGs for reasoning. On the other hand, removing the general LLM and relying solely on answers predicted by KG-specialized LLM leads to a noticeable decrease in precision, due to noises in its predictions. This highlighting the necessity of the general LLM for conducting inductive reasoning over multiple paths to derive final answers.

**Different LLMs.** We further analyze LLMs used for KG-specialized and general LLMs in Table 4. For KG-specialized LLMs, we directly plug the KG-Trie into different LLMs to conduct graph-constrained decoding and use Chat-GPT as the general LLM for final reasoning. For general LLMs, we adopt the same reasoning paths generated by KG-specialized LLMs to different LLMs to produce final answers. For zero-shot and few-shot learning, we adopt the original LLMs without fine-tuning, whose prompt templates can be found in Figures 9 and 11.

Table 4: Comparison of different LLMs used in GCR on WebQSP.

| Components | Learning Types | Variants | Hit | F1 |
|---|---|---|---|---|
| KG-specialized LLM | Zero-shot | Llama-3.1-8B | 28.25 | 10.32 |
| | | Llama-3.1-70B | 38.53 | 12.53 |
| | Few-shot | Llama-3.1-8B | 33.24 | 11.19 |
| | | Llama-3.1-70B | 41.13 | 13.14 |
| | Fine-tuned | Qwen2-0.5B | 87.48 | 60.03 |
| | | Qwen2-1.5B | 89.21 | 62.97 |
| | | Qwen2-7B | 92.31 | 72.74 |
| | | Llama-2-7B | 92.55 | **73.23** |
| | | Llama-3.1-8B | **92.74** | 73.14 |
| General LLM | Zero-shot | Qwen-2-7B | 86.32 | 67.59 |
| | | Llama-3.1-8B | 90.24 | 71.19 |
| | | Llama-3.1-70B | 90.24 | 71.19 |
| | | ChatGPT | **92.55** | 73.23 |
| | | GPT-4o-mini | 92.23 | **74.05** |

Results in Table 4 show that a lightweight LLM (0.5B) can outperform a large one (70B) after fine-tuning, indicating the effectiveness of fine-tuning in enhancing the ability of LLMs and make them specialized for KG reasoning. However, the larger LLMs (e.g., 7B and 8B) still perform better than smaller ones, highlighting the importance of model capacity in searching relevant reasoning paths

---

[1]Since there is no available code for EffiQA, we directly copy the results from the original paper.

on KGs. Similar trends are observed in general LLMs where larger models (e.g., GPT-4o-mini and ChatGPT) outperform smaller ones (e.g., Qwen-2-7B and Llama-3.1-8B), showcasing their stronger inductive reasoning abilities. This further emphasizes the need of paring powerful general LLMs with lightweight KG-specialized LLMs to achieve better reasoning driven by both of them.

**Parameter Analysis.** We first analyze the impact of different beam sizes $K$ for graph-constrained decoding on the performance of GCR. We conduct the experiments on WebQSP with different beam sizes of 1, 3, 5, 10, and 20. The results are shown in Figure 4. We observe that the hit and recall of GCR increase with the beam size. Because, with a larger beam size, the LLMs can explore more reasoning paths and find the correct answers. However, the F1 score, peaks when the beam size is set to 10. This is because the beam size of 10 can provide a balance between the exploration and exploitation of the reasoning paths. When the beam size is set to 20, the performance drops due to the increased complexity of the search space, which may introduce noise and make the reasoning less

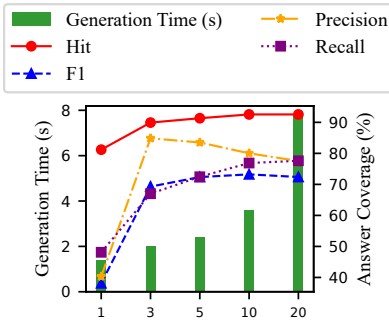

Figure 4: Parameter analysis of beam size $K$.

reliable. This also highlights the importance of using general LLMs to conduct inductive reasoning over multiple paths to disregard the noise and find the correct answers. Although the graph-constrained decoding benefits from the parallel GPU computation to explore multiple reasoning paths at the same time, the time cost still slightly increases from 1.4s to 7.8s with the increase of the beam size. Thus, we set the beam size to 10 in the experiments to balance the performance and efficiency. We also investigate the impact of $L$ hops paths used for KG-Trie construction in Appendix F.1. The results show that GCR can achieve a good balance between reasoning performance and efficiency by setting $L = 2$ and $K = 10$.

### 5.3 RQ2: HALLUCINATION ELIMINATION AND FAITHFUL REASONING

In this section, we investigate the effectiveness of KG constraints in eliminating hallucinations and ensuring faithful reasoning. We first compare the difference of answer accuracy (Hit) and *faithful reasoning ratio* by removing KG constraints in graph-constrained decoding. The faithful reasoning ratio is calculated as the percentage of faithful reasoning in *correctly predicted answers*. We define a reasoning as faithful where the generated reasoning path can be found in KGs, and vice versa.

From the Figure 5, we can observe that GCR achieves the 100% faithful reasoning ratio on both datasets, which indicates that GCR can eliminate hallucinations and ensure faithful reasoning during reasoning on KGs. In contrast, when removing KG constraints, both the answer accuracy and faithful reasoning decrease significantly on WebQSP. This shows that KG constraints not only improve reasoning by reducing the searching space, but also play a crucial role in preventing hallucinations for accurate reasoning. While the answer hit rate on CWQ remains almost unchanged, the ratio of faithful

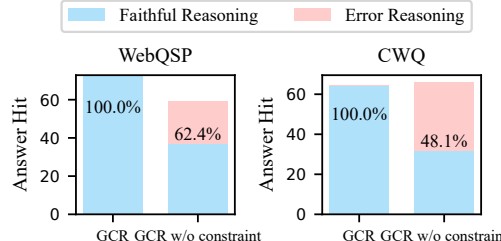

Figure 5: Analysis of performance and reasoning errors in GCR.

reasoning still decreases to 48.1%. This implies that even if LLMs can produce correct answers, the reasoning process is still prone to hallucinations and cannot be trusted, which is aligned with the findings in previous studies (Nguyen et al., 2024).

**Case Study.** We further provide a case study to illustrate the effectiveness of GCR in eliminating hallucinations and ensuring faithful reasoning. As shown in Table 5, the first case demonstrates that, without constraints, the model generates an incorrect reasoning path leading to an incorrect answer by hallucinating facts such as "Mabel Rose Ferguson is the child of Naill Ferguson and her parent is Alyssa Mastromonaco". In contrast, GCR generates a faithful reasoning path grounded in KGs that "Naill Ferguson has a child named Thomas Ferguson who has a parent named Ayaan Hirsi Ali". Based on the paths we can reason the correct answer to the question is "Ayaan Hirsi Ali". In the second case, although the LLM answers the question correctly, the generated reasoning path is still

Table 5: Examples of the faithful reasoning conducted by GCR. Red denotes the incorrect reasoning paths and answers, while **bold** denotes the correct paths and answers.

| **Case 1**: Incorrect answers and hallucinated reasoning paths without constraints. | |
| --- | --- |
| Question | Who is niall ferguson 's wife? |
| Answer | Ayaan Hirsi Ali |
| GCR $w/o$ constraint | # Reasoning Path: Niall Ferguson → people.person.children → Mabel Rose Ferguson → people.person.parents → Alyssa Mastromonaco
#Answer: Alyssa Mastromonaco |
| GCR | # Reasoning Path: Niall Ferguson → people.person.children → Thomas Ferguson → people.person.parents → **Ayaan Hirsi Ali**
#Answer: **Ayaan Hirsi Ali** |
| **Case 2**: Correct answers but hallucinated reasoning paths without constraints. | |
| Question | Where is jamarcus russell from? |
| Answer | Mobile |
| GCR $w/o$ constraint | # Reasoning Path: JaMarcus Russell → people.person.place_of_birth → Tampa
#Answer: **Mobile, Alabama** |
| GCR | # Reasoning Path: JaMarcus Russell → people.person.place_of_birth → **Mobile**
#Answer: **Mobile** |

hallucinated with incorrect facts. Conversely, GCR conducts faithful reasoning with both correct answer and reasoning path. These results demonstrate that GCR can effectively eliminate hallucinations and ensure faithful reasoning by leveraging KG constraints in graph-constrained decoding.

## 5.4 RQ3: Zero-shot Generalizability to Unseen KGs

In GCR, the knowledge graph is converted into a constraint which is plugged into the decoding process of LLMs. This allows GCR to generalize to unseen KGs without further training. To evaluate the generalizability of GCR, we conduct zero-shot transfer experiments on three unseen KGQA datasets: FreebaseQA (Jiang et al., 2019), CSQA (Talmor et al., 2019) and MedQA (Jin et al., 2021). Specifically, we use the same KG-specialized LLM (Llama-3.1-8B) trained on Freebase as well as two general LLMs (ChatGP, GPT-4o-mini). During reasoning, we directly plug the KG-Trie constructed from Freebase, ConceptNet and medical KGs into the GCR to conduct graph-constrained decoding without additional fine-tuning. The results are shown in Table 6.

Table 6: Zero-shot transferability to other KGQA datasets.

| Model | FreebaseQA | CSQA | MedQA |
| --- | --- | --- | --- |
| ChatGPT | 85 | 79 | 64 |
| GCR (ChatGPT) | **92** | **85** | **66** |
| GPT-4o-mini | 89 | 91 | 75 |
| GCR (GPT-4o-mini) | **94** | **94** | **79** |

From the results, it is evident that GCR outperforms ChatGPT and GPT-4o-mini in zero-shot performance on both datasets. Specifically, GCR shows 8.2% and 7.6% increase in accuracy on FreebaseQA and CSQA, respectively. This highlights the strong zero-shot generalizability of its graph reasoning capabilities to unseen datasets and KGs without additional training. However, the improvement on MedQA is not as significant as that on CSQA. We hypothesize this difference may be due to LLMs having more common sense knowledge, which aids in reasoning on common sense knowledge graphs effectively. On the other hand, medical KGs are more specialized and require domain-specific knowledge for reasoning, potentially limiting the generalizability of our method.

## 6 Conclusion

In this paper, we introduce a novel LLM reasoning paradigm called graph-constrained reasoning (GCR) to eliminate hallucination and ensure faithful reasoning by incorporating structured KGs. To bridge the unstructured reasoning in LLMs with the structured knowledge in KGs, we propose a KG-Trie to encode paths in KGs using a trie-based index. KG-Trie constrains the decoding process to guide a KG-specialized LLM to generate faithful reasoning paths grounded in KGs. By imposing constraints, we can not only eliminate hallucination in reasoning but also reduce the reasoning complexity, contributing to more efficient and accurate reasoning. Last, a powerful general LLM is utilized as a complement to inductively reason over multiple reasoning paths to generate the final answer. Extensive experiments demonstrate that GCR excels in faithful reasoning and generalizes well to reason on new KGs without additional fine-tuning.

## ETHICS STATEMENT

Our research focuses exclusively on scientific questions, with no involvement of human subjects, animals, or environmentally sensitive materials. Therefore, we foresee no ethical risks or conflicts of interest. We are committed to maintaining the highest standards of scientific integrity and ethics to ensure the validity and reliability of our findings.

## REPRODUCIBILITY STATEMENT

Our model is clearly formalized in the main text for clarity and comprehensive understanding. Detailed implementation, including dataset information, baselines, experimental settings, and model configurations, is provided in Appendices C to E. The experimental settings and baselines have been rigorously checked for fair comparison. Code and fine-tuned model weights will be made public upon acceptance.

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

# Appendix

## Table of Contents

## A  DETAILED RELATED WORK ON KG-ENHANCED LLMS

Knowledge graph (KG), as a structured representation of factual knowledge, has been widely used to enhance the factual knowledge and reasoning abilities of LLMs (Pan et al., 2024) by reducing the hallucinations (Nguyen et al., 2024; Dhuliawala et al.; Lv et al., 2024). In this section, we provide a detailed review of the related work on KG-enhanced LLMs, which can be categorized into two paradigms: retrieval-based and agent-based methods.

**Retrieval-based Methods.** Retrieval-based methods retrieve relevant facts from KGs with an external retriever and then feed them into the inputs of LLMs for reasoning. These methods aim to provide LLMs with external knowledge to enhance their reasoning abilities. For example, KD-CoT (Wang et al., 2023) retrieves relevant knowledge from KGs to generate faithful reasoning plans for LLMs. EWEK-QA (Dehghan et al., 2024) enriches the retrieved knowledge by searching from both KGs and the web. RoG (Luo et al., 2024) proposes a planning-retrieval-reasoning framework that retrieves reasoning paths from KGs to guide LLMs conducting faithful reasoning. GNN-RAG (Mavromatis & Karypis, 2024) adopts a lightweight graph neural network to effectively retrieve from KGs. GNN-RAG+RA (Mavromatis & Karypis, 2024) combines the retrieval results of both RoG and GNN-RAG to enhance the reasoning performance. However, these methods may suffer from the retrieval accuracy, which limits the reasoning performance.

**Agent-based Methods.** Agent-based methods treat LLMs as agents that iteratively interact with KGs to find reasoning paths and answers. For example, StructGPT (Jiang et al., 2023) treats LLMs as agents to interact with KGs to find a reasoning path leading to the correct answer. ToG (Sun et al., 2024) extends the method and conducts reasoning on KGs by exploring multiple paths and concludes the final answer by aggregating the evidence from them. EffiQA (Jiang et al., 2024) proposes an efficient agent-based method to reason on KGs. Plan-on-Graph (Chen et al.) proposes an adaptive planing paradigm to decompose the question into sub-tasks and guide the LLMs to reason on KGs. Debate on Graph (Ma et al., 2024) asks LLM as agents to debate with each other to gradually simplify complex questions and find the correct answers. Although these methods are effective, they face high computational costs and challenges in designing the interaction process.

## B  KG-TRIE CONSTRUCTION

KG-Trie converts KG structures into the format that LLMs can handle. It can been incorporated into the LLM decoding process as constraints, allowing for faithful reasoning paths that align with the graph's structure. The KG-Trie can be either pre-computed for fast inference or constructed on-demand to minimize pre-processing time.

### B.1 CONSTRUCTION STRATEGIES

**Offline Construction.** The KG-Trie can be pre-computed offline, allowing them to be used during inference at no additional cost. Instead of constructing the KG-Trie for all entities in the KG, we could only construct the KG-Trie for certain entities. We can select the entities based on their popularity, importance, or the frequency of their occurrence in the questions.

**On-demand Construction.** Alternatively, we can construct the KG-Trie on-demand. When a question is given, we first identify the question entities with named entity recognition (NER) tools. Then, we retrieve the question-related subgraphs around the question entities from the KGs. Finally, we construct a question-specific KG-Trie based on the retrieved subgraphs. The KG-Trie is then used to guide the LLMs to reason on the KGs.

**Dynamic Cache for KG-Trie Construction.** Users can also develop their own strategies to balance pre-processing and inference overhead. For example, we can maintain a dynamic cache to store the KG-Trie for the most frequently asked questions, as shown in Figure 6. When a new question is given, they first check whether the KG-Trie for the question is in the cache. If it is, they directly use the KG-Trie for inference. Otherwise, they construct a question-specific KG-Trie on-demand. The cache can be updated periodically to remove the least frequently used KG-Trie and add the new ones.

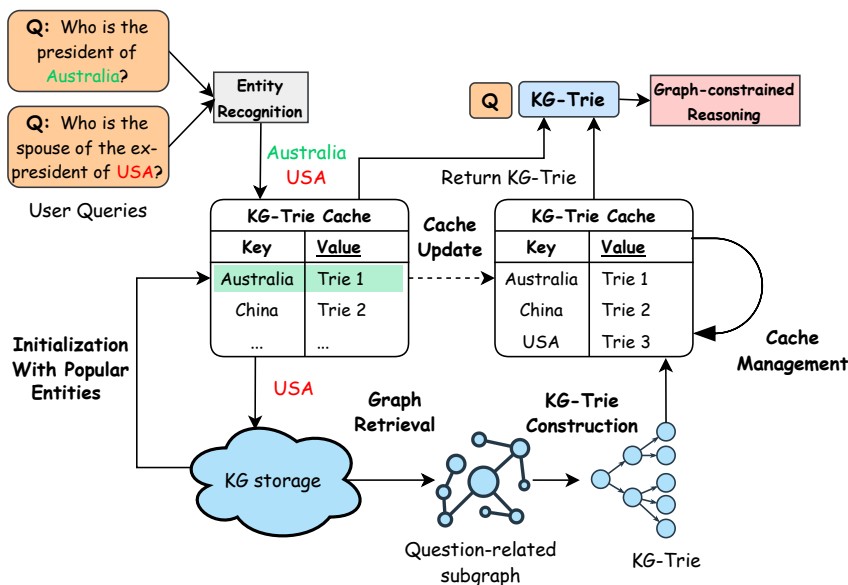

Figure 6: The illustration of dynamic cache for KG-Trie construction.

### B.2 TIME AND SPACE COMPLEXITY ANALYSIS

The time and space complexity for KG-Trie construction is affordable and can be easily improved in industry-level applications to support billions of scale graphs. To support this, we provide detailed theoretical analysis and empirical evidence. In experiments, we adopt the breadth-first search, whose complexities are:

#### B.2.1 THEORETICAL ANALYSIS

**Time Complexity.** Constructing the KG-Trie involves a BFS traversal to explore paths up to a maximum length of $L$ starting from certain entities. The time complexity of this traversal is $O(E^L)$, where $E$ is the average number of edges per entity, and $L$ is the maximum path length. BFS ensures that all reachable paths up to length $L$ are considered. However, BFS can be replaced with other efficient graph-traversing algorithms, such as random walk (Xia et al., 2019) to further improve efficiency.

**Space Complexity.** The space complexity of the KG-Trie depends on the number of unique paths and their tokenized representations. In the worst case, the space complexity is $O(E^L \times T)$, where $T$ represents the average number of tokens per path. Trie structures are efficient for storing shared prefixes, which reduces redundancy and optimizes memory usage. Moreover, it supports efficient traversal of reasoning paths in constant time.

### B.2.2 EMPIRICAL ANALYSIS

We have provided the average BFS running time and space consumption of the KG-Trie construction to demonstrate its efficiency. The system settings are illustrated at Table 7.

Table 7: System settings overview for efficiency experiments.

| System Setting | Specification |
|---|---|
| CPU | Intel(R) Xeon(R) Silver 4214R CPU @ 2.40GHz |
| Memory | 32G |
| BFS Implementation | Virtuoso SPARQL |
| Space Storage | Pickle |

In the experiment, we build the KG-Trie for all question entities of WebQSP dataset and measure the average running time and space consumption. The BFS is executed on the Freebase KG stored in a Virtuoso database (Erling & Mikhailov, 2009). We retrieve the $L$-hop paths, then save the constructed KG-Trie with Pickle. The statistics show that both running time and space usage are acceptable when $L <= 3$, which highlights efficiency in KG-Trie construction. Although a larger hop can lead to better coverage of the possible answer, it would significantly increase the time and space complexity. Thus, we set hops to 2 or 3 in experiments to balance between efficiency and effectiveness. Notably, time can be further reduced by utilizing multi-threading. Space consumption can be optimized by storing data in a database.

### B.3 STRATEGIES FOR OPTIMIZING EFFICIENCY

We provide several strategies that can be used to further speed up the KG-Trie construction.

**Parallel Processing:** As the KG-Trie is independently constructed for each entity, it can be easily scaled with parallel processing. We provide the total running time of constructing 2-hop KG-Trie of all question entities in WebQSP dataset in Table 9 to show the improvement of parallel processing. It shows that the efficiency can be greatly improved with parallel processing. This parallel nature enables it to be executed on distributed computing systems such as Hadoop and Spark in real-world applications.

**Efficiency Graph Traversal Algorithms:** The BFS or DFS enumerates all the paths around the entities which might lead to computational overhead. However, they can be easily replaced with other graph traversal algorithms, such as random walk, to reduce time complexity.

**Combination with Graph Retrieval Algorithms:** To reduce the overhead of graph traversal, we can construct the KG-Trie on the question-related subgraphs. To this end, our methods can be combined with other graph retrieval algorithms, such as GNN-RAG (Mavromatis & Karypis, 2024) and RoG (Luo et al., 2024). They would retrieve meaningful and relevant paths from KGs to speed up the KG-Trie construction. However, the performance might be limited by the retrieval accuracy.

Table 8: Average running time and space utilization of the KG-Trie construction.

| Hop | Avg. Running Time (s) | Space (Mb) |
|---|---|---|
| L=1 | 0.0058 | 0.4 |
| L=2 | 0.0133 | 0.5 |
| L=3 | 0.0219 | 2.5 |

Table 9: Total running time and improvement under different processing threads.

| Total time (s) | Total Time (min) | Improvement |
|---|---|---|
| Thread=1 | 4.03 | 100% |
| Thread=4 | 3.21 | 126% |
| Thread=10 | 2.31 | 174% |
| Thread=20 | 1.92 | 210% |

**Reduce Entities Number:** Instead of constructing the KG-Trie for all entities in the KG, we could only construct the KG-Trie for certain entities. We can select the entities based on their popularity, importance, or the frequency of their occurrence in the questions.

### B.4 REAL-WORLD APPLICABILITY

To support real-world applications with billion-scale KGs, KG-Trie construction can be implemented in industrial-level settings. For instance, billion-scale KGs can be stored in scalable graph databases like Neo4j. The parallel nature of KG-Trie construction allows it to be executed on distributed computing systems such as Hadoop and Spark, enabling pre-computation and offline storage. The constructed KG-Trie can then be stored in a database and loaded for inference without additional computation, facilitating real-time responses. To reduce the overhead in pre-processing, we can design a cache mechanism that only builds KG-Trie for popular accessed entities and caches them for faster inference. The illustration of the framework can be found in Figure 6.

## C DATASETS

**KGQA Datasets.** To compare the reasoning performance with existing methods, we use two benchmark KGQA datasets in this study: WebQuestionSP (WebQSP) (Yih et al., 2016) and Complex WebQuestions (CWQ) (Talmor & Berant, 2018). To ensure fairness, we adopt the same train and test splits as previous works (Jiang et al., 2022; Luo et al., 2024). Details of the datasets can be found in Table 10.

Both WebQSP and CWQ can be reasoned using Freebase KGs[2] (Bollacker et al., 2008). To reduce the size of the KGs, we use a subgraph of Freebase by extracting all triples that start from question entities within the maximum reasoning hops provided by previous works[3] (Luo et al., 2024). The statistics of the knowledge graphs are shown in Table 12.

**Fine-tuning Datasets.** To enhance the KG reasoning ability of LLMs, we construct fine-tuning datasets by generating reasoning paths from the KGs. Specifically, we adopt the training split of WebQSP and CWQ, which contain 2,826 and 27,639 question-answer pairs, respectively. For each question, we find all the shortest reasoning paths on KGs that connect the question entity to the answer entity. We then convert the reasoning paths into formatted strings and pair them with the question-answer pairs with the template shown in Figure 9 to form the fine-tuning datasets. Since there could be multiple reasoning paths for a question, we generate multiple training instances paired with different reasoning paths for each question-answer pair. The fine-tuning datasets contain 28,307 and 181,602 question-reasoning path-answer triples for WebQSP and CWQ, respectively. The statistics of the fine-tuning datasets are shown in Table 11.

**Zero-shot Generalization Datasets.** To evaluate the transferability of GCR, we further select three new KGQA datasets: FreebaseQA (Jiang et al., 2019), CommonsenseQA (CSQA) (Talmor et al., 2019) and MedQA-USMLE (MedQA) (Jin et al., 2021).FreebaseQA is an open-ended question answering dataset. CSQA is a 5-way multiple choice QA dataset that involves reasoning with commonsense knowledge. MedQA is a 4-way multiple choice QA task that requires biomedical and clinical knowledge. FreebaseQA adopts the same Freebase KG used in WebQSP and CWQ. For CSQA, we use the ConceptNet (Speer et al., 2017), which is a general-purpose KG that contains

---

[2]`https://github.com/microsoft/FastRDFStore`

[3]WebQSP: `https://huggingface.co/datasets/rmanluo/RoG-webqsp`, CWQ: `https://huggingface.co/datasets/rmanluo/RoG-cwq`

commonsense knowledge. For MedQA, we use a medical KG constructed from the Unified Medical Language System (Yasunaga et al., 2021). The statistics of the knowledge graphs are shown in Table 12. We respectively select 100 questions from each dataset. For each question, following previous studies (Feng et al., 2020; Yasunaga et al., 2021), a 2-hop subgraph is extracted from the KGs to form the zero-shot generalization datasets.

Table 10: Statistics of datasets.

| Dataset | Dataset Statistics | | Statistics of Answer Numbers | | | |
|---|---|---|---|---|---|---|
| | #Train | #Test | #Ans = 1 | $2 \geq$ #Ans $\leq 4$ | $5 \geq$ #Ans $\leq 9$ | #Ans $\geq 10$ |
| WebQSP | 2,826 | 1,628 | 51.2% | 27.4% | 8.3% | 12.1% |
| CWQ | 27,639 | 3,531 | 70.6% | 19.4% | 6% | 4% |

Table 11: Statistics of fine-tuning datasets for graph-constrained decoding.

| Total | WebQSP | CWQ |
|---|---|---|
| 209,909 | 28,307 | 181,602 |

Table 12: Statistics of constructed knowledge graphs.

| KG | #Entities | #Relations | #Triples |
|---|---|---|---|
| Freebase | 2,566,291 | 7,058 | 8,309,195 |
| ConceptNet | 799,273 | 17 | 2,151,303 |
| MedKG | 9,958 | 15 | 49,974 |

# D BASELINES

We compare GCR with the 22 baselines grouped into three categories: 1) *LLM reasoning methods*, 2) *graph reasoning methods*, and 3) *KG-enhanced LLM reasoning methods*. The details of each baseline are described as follows.

**LLM reasoning methods** only rely on LLMs for reasoning without utilizing external KGs. We include both the vanilla LLMs with different sizes and the LLMs with advanced reasoning mechanisms. Specifically, we consider the following baselines:

- Qwen2-0.5B/1.5B.7B (Yang et al., 2024a) provides a series of pre-trained LLMs with different sizes, including 0.5B, 1.5B, and 7B parameters.
- Llama-2-7B (Touvron et al., 2023) is a large-scale LLM pre-trained on a diverse range of tasks.
- Llama-3.1-8B (Meta, 2024) is the updated version of Llama-2 with more powerful reasoning capabilities.
- ChatGPT (OpenAI, 2022) is a powerful closed-source LLM that could follow instructions to conduct complex tasks.
- GPT-4o-mini (OpenAI, 2024a) is the new flagship model of OpenAI that could reason across different modalities and tasks.
- Few-shot prompt (Brown et al., 2020) is a few-shot learning method that provides LLMs with a few examples in the prompts to conduct reasoning.
- CoT (Wei et al., 2022) is a chain-of-thought reasoning method that prompts LLMs to generate a chain of reasoning steps.
- Self-consistency (Wang et al., 2024) generates multiple reasoning paths and selects the most consistent answer.

**Graph reasoning methods** focus on reasoning on KGs using graph neural networks (GNNs) (Wu et al., 2020) or graph-based reasoning mechanisms. We include the following baselines:

- GraftNet (Sun et al., 2018) is a graph-based reasoning method that retrieves relevant subgraphs from KGs with entity linking.
- NSM (He et al., 2021) utilizes the sequential model to mimic the multi-hop reasoning process on KGs.
- SR+NSM (Zhang et al., 2022) proposes a relation-path retrieval to retrieve subgraphs for multi-hop reasoning.
- ReaRev (Mavromatis & Karypis, 2022) is a GNN-based method that reasons on KGs by considering complex graph information.
- UniKGQA (Jiang et al., 2022) is a unified framework that combines graph-based reasoning of GNNs and LLMs for KGQA.

**KG-enhanced LLM reasoning methods** incorporate KGs to enhance the reasoning abilities of LLMs which can be further divided into retrieval-based and agent-based paradigms. We include the following baselines:

*Retrieval-based methods* retrieve relevant facts from KGs with an external retriever and then feed them into the inputs of LLMs for reasoning:

- KD-CoT (Wang et al., 2023) retrieves relevant knowledge from KGs to generate faithful reasoning plans for LLMs.
- EWEK-QA (Dehghan et al., 2024) enriches the retrieved knowledge by searching from both KGs and web.
- RoG (Luo et al., 2024) proposes a planning-retrieval-reasoning framework that retrieves reasoning paths from KGs to guide LLMs conducting faithful reasoning.
- GNN-RAG (Mavromatis & Karypis, 2024) adopts a lightweight graph neural network to effectively retrieve from KGs.
- GNN-RAG+RA (Mavromatis & Karypis, 2024) combines the retrieval results of both RoG and GNN-RAG to enhance the reasoning performance.

*Agent-based methods* treat LLMs as agents that iteratively interact with KGs to find reasoning paths and answers:

- ToG (Sun et al., 2024) conducts the reasoning on KGs by exploring multiple paths and concludes the final answer by aggregating the evidence from them.
- EffiQA (Jiang et al., 2024) proposes an efficient agent-based method to reason on KGs.

## E  IMPLEMENTATION DETAILS AND EXPERIMENT SETTINGS

In this section, we will detail the implementation of GCR as well as the experiment settings.

**Fine-tuning KG-specialized LLMs.** We fine-tune several lightweight LLMs ranging from 0.5B to 8B (Yang et al., 2024a; Touvron et al., 2023; Meta, 2024) on the fine-tuning datasets for 3 epochs. The batch size is set to 4 and the learning rate is set to 2e-5. We use the cosine learning rate scheduler policy with the warmup ratio set to 0.03. The training is conducted on 2 A100-80G GPUs for each model. The training time and memory usage are shown in Table 13.

**KGQA Experiment Settings.** The KGQA experiment shown in Table 1 aims to compare the reasoning performance of GCR with existing methods. For our method, we use the fine-tuned Llama-3.1-8B as KG-specialized LLMs, the general LLM is selected as ChatGPT and GPT-4o-mini. The KG-Trie is constructed from the subgraph of Freebase KGs. The maximum reasoning hops are set to 2 for both WebQSP and CWQ. The beam size is set to 10 for graph-constrained decoding. For vanilla LLMs baselines, we use the zero-shot prompting to ask the models to answer the questions. For other baselines, we strictly check whether the original papers follow the same settings and copy the results for fair comparison.

Table 13: Training time and memory usage for different KG-specialized LLMs.

| Model | Time | Mem. Usage per GPU |
|-------|------|--------------------|
| Qwen2-0.5B | 3.47h | 10G |
| Qwen2-1.5B | 4.11h | 25G |
| Qwen2-7B | 14.37h | 81G |
| Llama-2-7B | 13.93h | 80G |
| Llama-3.1-8B | 14.52h | 85G |

**Efficiency Analysis Settings.** The efficiency analysis shown in Table 2 aims to compare the efficiency and performance of different methods on WebQSP. For GCR, we use the same settings as the KGQA experiment. For dense retriever methods (e.g., S-Bert (Reimers & Gurevych, 2019), BGE (Zhang et al., 2023), OpenAI-Emb. (OpenAI, 2024b)), we first search all paths within 2-hops on the KGs which are formatted as sentences with the template in Figure 8. Then, we adopt the embedding model to encode the path sentences as embeddings which are stored in a vector database. During inference, we retrieve 10 paths from the vector database with the question as query and feed them into the LLMs for reasoning. For GNN-RAG (Mavromatis & Karypis, 2024) and RoG (Luo et al., 2024), we strictly follow the original papers to retrieve reasoning paths and conduct the experiments. For agent-based methods (e.g., ToG (Sun et al., 2024)), we use the same settings detailed in the original papers. For EffiQA (Jiang et al., 2024), since there is no available code, we directly copy the results from the original paper.

The average runtime is measured by the time taken to answer the questions. The average number of LLM calls is the number of times the LLMs are called to answer the questions. The average number of LLM tokens is the number of tokens inputted into LLMs to answer the questions, such as questions and retrieved reasoning paths. The experiments are conducted on a single A100-80G GPU for each method.

**Ablation Study.** In ablation study, we first try to analyze the effectiveness of different components in GCR. We conduct the experiments on WebQSP and CWQ datasets. By removing the KG-specialized LLM ($w/o$ KG-specialized LLM), we search all the 2-hop paths starting from question entities and feed them into the general LLMs for reasoning. By removing the general LLM ($w/o$ general LLM), we directly use the hypothesis answers generated by the KG-specialized LLMs as the final answers.

**Different LLMs.** We also analyze the different LLMs used for KG-specialized LLMs and general LLMs on WebQSP. For KG-specialized LLMs, we first use the vanilla LLMs with different learning types (i.e., zero-shot and few-shot prompting). For zero-shot prompting, we directly ask the models to generate the reasoning paths with the constraints. For few-shot prompting, we provide the models with a few examples in the prompts to conduct path generation. Detailed prompts can be found in Figures 9 and 11. Then, we fine-tune the lightweight LLMs with different sizes (0.5B to 8B) on the graph-constrained decoding task. For general LLMs, we use the vanilla LLMs to directly conduct reasoning over multiple reasoning paths. The detailed reasoning prompts can be found in Figure 10.

**Parameter Analysis.** We first analyze the performance of GCR with different beam sizes for graph-constrained decoding. We conduct the experiments on the WebQSP datasets with beam sizes of 1, 3, 5, 10, and 20. Then, we analyze the performance of GCR with different hops of paths encoded in the KG-Trie. We conduct the experiments on the WebQSP datasets with maximum paths hops ranging from 1 to 4.

**Faithful Reasoning Analysis.** We investigate the effect of the KG constraints on ensuring faithful reasoning. We adopt the fine-tuned Llama-3.1-8B as KG-specialized LLMs. Then, we compare the faithful reasoning rate and answer hit of GCR with and without the KG constraints in graph-constrained decoding. The faithful reasoning rate is the percentage of the faithful reasoning in the correctly predicted answers. A reasoning path is considered faithful if it can be found in the KGs, and vice versa. The answer hit is the percentage of the correct answers in the predictions.

**Zero-shot Generalization Analysis.** We evaluate the transferability of GCR on two zero-shot generalization datasets: CSQA and MedQA. We use the fine-tuned Llama-3.1-8B as KG-specialized LLMs and ChatGPT as well as GPT-4o-mini as the general LLMs. The KG-Trie is constructed from the subgraph of ConceptNet and MedKG. The maximum reasoning hops are set to 2 for both

datasets. The beam size is set to 10 for graph-constrained decoding. For vanilla LLMs baselines (i.e., ChatGPT and GPT-4o-mini), we use the zero-shot prompting to ask the models to answer the questions.

## F ADDITIONAL EXPERIMENT RESULTS

### F.1 PERFORMANCE ON DIFFERENT HOPS OF KG-TRIE

In this section, we analyze the impact of different hops of reasoning paths on the performance of GCR. We conduct the experiments on WebQSP with different maximum hops of reasoning paths encoded in the KG-Trie. The results are shown in Figure 7. We observe that the performance of GCR increases with the number of hops of reasoning paths. The performance peaks when the maximum hops of reasoning paths are set to 2. This is because the 2-hop paths can provide sufficient information for the LLMs to conduct reasoning. When the hops are set to 3 or 4, the performance drops due to the increased complexity of the reasoning paths, which may introduce noise and make the reasoning less reliable. Additionally, the size of the KG-Trie slightly increases from 0.5 MB to 7.5 MB with the increase of the hops from 1 to 4. This indicates that the KG-Trie can be efficiently constructed with a small size and guide the LLMs to reason on graphs effectively.

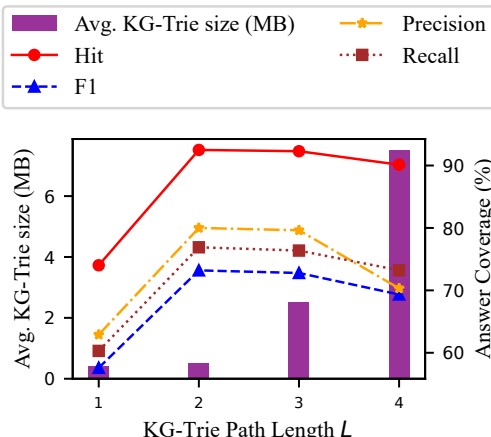

Figure 7: Parameter analysis of path hop $L$ for KG-Trie construction on WebQSP.

### F.2 PERFORMANCE ON MULTI-PATH REASONING

GCR could take advantage of the GPU parallel computation to conduct multi-path explorations on KGs with beam-search. It could generate simultaneously generate $K$ reasoning paths and hypothesis answers with beam search in a single LLM call. The effectiveness of different $K$ is analyzed in Figure 4 where larger $K$ can lead to a better recall of the answers. In addition, we compare the F1 performance under different numbers of ground-truth answers with RoG, which requires reasoning across multiple reasoning paths to find all answers. From the results shown in Table 14, we can observe that GCR exhibits better performance in exploring multiple paths for reasoning.

Table 14: F1 comparison against RoG under different numbers of ground-truth answers.

| Methods | WebQSP | | | | CWQ | | | |
|---|---|---|---|---|---|---|---|---|
| | # Ans = 1 | 2 <= # Ans <= 4 | 5 <= # Ans <= 9 | # Ans >= 10 | # Ans = 1 | 2 <= # Ans <= 4 | 5 <= # Ans <= 9 | # Ans >= 10 |
| GCR | **71.31** | 78.14 | **83.47** | **63.20** | 55.80 | **64.08** | **62.57** | **55.32** |
| RoG | 67.89 | **79.39** | 75.04 | 58.33 | **56.9** | 53.73 | 58.36 | 43.62 |

## F.3 PERFORMANCE ON MULTI-HOP REASONING

To demonstrate the effectiveness of multi-hop reasonings. We illustrate the F1 performance under different hops. From results shown in Table 15, we can observe that GCR also outperforms baselines in multi-hop reasoning.

Table 15: F1 comparison against RoG under different hops of reasoning.

| Methods | WebQSP | | | CWQ | | |
|---|---|---|---|---|---|---|
| | 1 hop | 2 hop | >=3 hop | 1 hop | 2 hop | >=3 hop |
| GCR | 75.05 | **72.72** | - | **64.54** | **62.44** | **43.82** |
| RoG | **77.03** | 64.86 | - | 62.88 | 58.46 | 37.82 |

## F.4 ANALYSIS OF THE FAILURE CASES

Although GCR achieves 100% trustful reasoning, there are still some failure cases due to the noise and redundant information in KGs. Two failure cases are presented in Table 16. In the first case, the generated path is unrelated to the question. GCR provides a valid reasoning path that describes Anna Bligh's political position, which lacks information about her electoral district. Although LLMs exhibit strong reasoning ability, they still cannot always find meaningful paths, resulting in incorrect answers. In the second case, the KG is incomplete, and the generated path does not contain facts for generating answers. Although KGs store abundant factual knowledge, there are still missing facts. Because there is no information about the character's player stored in KGs, GCR cannot generate the correct answer. These failure cases indicate that the performance of GCR can be further improved by enhancing the reasoning ability of LLMs and the completeness of KGs.

Table 16: Failure cases predicted by GCR.

| **Case 1**: Generated paths are unrelated to the questions. | |
|---|---|
| Question | What electorate does anna bligh representt? |
| Answer | Electoral district of South Brisbane |
| Generated Path | Anna Bligh → government.politician.government_positions_held → m.0cr320w → government.government_position_held.jurisdiction_of_office → Queensland |
| Predicted answer | Queensland |
| **Case 2**: KG incompleteness. | |
| Question | who plays ken barlow in coronation street? |
| Answer | William Roache |
| Generated Path | Coronation Street → tv.tv_program.program_creator → Tony Warren → fictional_universe.fictional_character_creator.fictional_characters_created → Ken Barlow |
| Predicted answer | Ken Barlow |

# G LIMITATIONS

In this section, we discuss the limitations and future directions of the proposed method.

- **Definition of Zero-hallucination.** This paper defines KG-constrained zero-hallucination as the generated reasoning paths are fully grounded in the KG. However, KGs often face issues of incompleteness and incorrect facts, leading to occasional false positives. Detecting such hallucinations without external evidence remains challenging, highlighting the potential of integrating cross-references from multiple knowledge sources—such as KGs, web data, and documents—to improve reasoning faithfulness.

- **Time Complexity of Complex Questions.** Highly complex questions usually require conduct reasoning with multiple steps. However, directly constructing a KG-Trie for a larger

*L* can be time-consuming. To address this, GCR can be integrated with existing planning-based methods to decompose complex questions into multiple shorter steps (Li et al., 2024). By breaking down the reasoning process, we can construct a KG-Trie with a smaller *L* for each subtask to conduct reasoning, thereby reducing computational overhead while maintaining inference quality.

- **Irrelevant Reasoning Path.** As shown in Appendix F.4, although LLMs exhibit strong reasoning ability, they still cannot always find meaningful paths, resulting in incorrect answers. It is worth to investigate how to further improve the reasoning ability of LLMs, especially under the settings of incomplete knowledge graphs.

## H  TEMPLATES AND PROMPTS

In this section, we illustrate all the templates and prompts used in the experiments.

**Path Sentence Template.** The template for converting reasoning paths into natural language sentences is shown in Figure 8, where the $e_*$ and $r_*$ denotes the entities and relations in a reasoning path $\boldsymbol{w_z} = e_0 \xrightarrow{r_1} e_1 \xrightarrow{r_2} \ldots \xrightarrow{r_l} e_l$,

---

**Path Sentence Template**

`<PATH>` $e_1 \rightarrow r_1 \rightarrow e_2 \rightarrow \ldots \rightarrow r_l \rightarrow e_l$ `</PATH>`

---

Figure 8: The template for converting reasoning paths into formatted sentences.

**Graph-constrained Decoding Prompt.** The prompt for graph-constrained decoding is shown in Figure 9, where the question and mentioned entities are provided to the LLMs to generate reasoning paths and hypothesis answers. In the fine-tuning datasets, the supervised LLM outputs are constructed from the ground-truth answers and reasoning paths extracted from the KGs.

---

**Graph-constrained Decoding Prompt**

=========================== Prompt Input ===========================
Reasoning path is a sequence of triples in the KG that connects the topic entities in the question to answer entities. Given a question, please generate some reasoning paths in the KG starting from the topic entities to answer the question.

# Question:
`<Question>`

# Topic entities:
`<Question Entities>`

=========================== LLM Output ===========================
# Reasoning Path:
`<PATH> <Reasoning Path> </PATH>`

# Answer:
`<Hypothesis Answer>`

---

Figure 9: The prompt template for graph-constrained decoding.

The few-shot prompt template for graph-constrained decoding is shown in Figure 11. We provide a few examples in the prompts to guide the LLMs to generate reasoning paths. Since the LLMs with few-shot prompt learning are not fine-tuned on the graph-constrained decoding task, we only apply the constraint to generate reasoning paths.

**Graph Inductive Reasoning Prompt.** The prompt for graph inductive reasoning is shown in Figure 10. We adopt the graph-constrained decoding to generate $K$ reasoning paths and hypothesis answers for each question. The reasoning paths and hypothesis answers are provided to the general LLMs to answer the questions without fine-tuning.

---

**Graph Inductive Reasoning Prompt**

============================ Prompt Input ===============================
# Reasoning Paths:
```
<Reasoning Path 1><Hypothesis Answer 1>
...
<Reasoning Path K><Hypothesis Answer K>
```

# Question:
```
<Question>
```

Based on the reasoning paths, please answer the given question. Please keep the answer as simple as possible and only return answers. Please return each answer in a new line.

============================ LLM Output ===============================
```
<Answer 1>
<Answer 2>
...
```

---

Figure 10: The prompt template for graph inductive reasoning.

### Few-shot Graph-constrained Decoding Prompt

```
============================ Prompt Input ==============================
Reasoning path is a sequence of triples in the KG that connects the topic entities in the question to
answer entities. Given a question, please generate some reasoning paths in the KG starting from the
topic entities to answer the question.

Example 1

# Question:
<Question>

# Topic entities:
<Question Entities>

# Reasoning Path:
<Reasoning Path>

Example 2

# Question:
<Question>

# Topic entities:
<Question Entities>

# Reasoning Path:
<Reasoning Path>

Example 3

# Question:
<Question>

# Topic entities:
<Question Entities>

# Reasoning Path:
<Reasoning Path>

Input

# Question:
<Question>

# Topic entities:
<Question Entities>

============================ LLM Output ==============================
# Reasoning Path:
<Reasoning Path>
```

Figure 11: The few-shot prompt template for graph-constrained decoding.

