# OpenReview forum: "Graph-constrained Reasoning: Faithful Reasoning on Knowledge Graphs with Large Language Models"
_ICLR.cc/2025/Conference — Submitted to ICLR 2025_

### Official Review · Reviewer_SPbz · 2024-10-17

**Soundness:** 2
**Presentation:** 2
**Contribution:** 2
**Rating:** 5
**Confidence:** 4

**Summary:**

This paper proposes a graph-constrained reasoning (GCR) framework to enable LLMs to produce faithful reasoning and reduce hallucinations. The key idea of GCR is to convert vanilla KG into a KG-trie and enable LLMs to perform graph-constrained decoding. Experiments on various KGQA reasoning benchmarks and several LLMs demonstrate the effectiveness of the proposed GCR.

**Strengths:**

1. The conversion of a KG into KG-trie and the introduction of graph-constrained decoding are reasonable.

2. Extensive experiments on the various KGQA benchmarks demonstrate the effectiveness of GCR.

3. The results of zero-shot generalizability for reasoning on unseen KGs are interesting.

**Weaknesses:**

1. What is the time cost of the construction of a KG-trie? Furthermore, since the construction of a KG-trie is query-dependent, the offline pre-constructed KG-trie strategy may not be effective when a new input query is introduced. Additionally, the beam search method for constructing a KG-trie may be time-consuming.

2. Real-world KGs are often incomplete and contaminated. Therefore, when entities are absent from the KG or when the reasoning paths generated by Figure 3 (Lines 220-228) contain unreliable logical pathways, how does GCR work? Will GCR still contribute positively to the results under these circumstances?

3. The results of the zero-shot generalizability improvement in Table 6 for MedQA are not particularly significant. I am curious about the additional time cost associated with implementing GCR compared to vanilla ChatGPT and GPT-4o-mini.

4. The authors may want to discuss or summarize more KG-enhanced methods for reducing knowledge  hallucinations including [A, B].

[A] Chain-of-Verification Reduces Hallucination in Large Language Models. ACL 2024 Findings.

[B] Coarse-to-Fine Highlighting: Reducing Knowledge Hallucination in Large Language Models. ICML2024.

**Questions:**

1. The reasoning path in a KG-trie may consist of multiple paths. In cases where several valid reasoning paths exist within the KG-trie, how does GCR operate?

2. Given that the knowledge stored in KGs often exhibits redundancy, can we ensure that all generated reasoning paths are strongly relevant to the question posed? Is there a risk of introducing irrelevant information?

3. In line 917, the beam size is set to 10 for graph-constrained decoding. Is the beam size sensitive to graph-constrained reasoning, and if so, what aspects of GCR are influenced by the beam size?

4. If a KG is temporal and dynamic, how can the pre-constructed KG-trie strategy be effectively employed?

---

> ### Author Response · Authors · 2024-11-18
> **Official response by authors to Reviewer SPbz: Part 1**
>
> Thank you for your detailed review and constructive feedback on our submission. We appreciate your recognition of the strengths of our work and your thoughtful comments on areas where we could improve. Below, we address your main concerns:
>
> ## Weakness 1: Computational Expense of KG-Trie Construction
>
> **We want to clarify that there is no need to build a KG-Trie for all entities in KGs.** In experiments, we only construct the KG-Trie for entities mentioned in questions. The KG-Trie can be either pre-computed or constructed on-demand to minimize pre-processing time. When the user’s questions are comping, we can identify the mentioned question entities and retrieve the question-related subgraphs from KGs for KG-Trie construction. This process is also very efficient, where the detailed analysis of time complexity and actual running time can be found in **our responses to all reviewers.** We also discuss the potential solutions to further improve the efficiency and scale into real-world applications with billion-scale KGs.
>
> ## Weakness 2: Adaptability to Real-World KG Incompleteness
>
> We thank you for the insightful comments. Facts in KGs are usually more clean and trustworthy than open-world knowledge, which could serve as a convincing source of knowledge to guide reasoning in GCR. However, the incompleteness of the KGs could still undermine the accuracy of the GCR. As shown in the error cases (response to weakness 3 of reviewer hR19), the missing knowledge would mislead the reasoning of GCR. We will explore the reasoning for incomplete KGs in the future.
>
> To alleviate the effects of unreliable paths, we propose the graph inductive reasoning module. We adopt a powerful general LLM to reason over multiple generated paths and select useful paths to produce final answers. For example,
>
> > **Question:** who did jackie robinson first play for?
> > **Ground-truth answer:** UCLA Bruins football
> > **Generated paths:**
> > Jackie Robinson \-\> sports.pro\_athlete.teams \-\> m.0hpgh\_h \-\> sports.sports\_team\_roster.team \-\> UCLA Bruins football
> > Jackie Robinson \-\> baseball.baseball\_player.batting\_stats \-\> m.06sbpz2 \-\> baseball.batting\_statistics.team \-\> Brooklyn Dodgers
> > **Predicted answer:** UCLA Bruins football
>
> In this example, two reasoning paths about the team of Jackie Robinson are generated. However, only the first one is the first team of Jackie Robinson. Thus, based on the internal knowledge of powerful LLMs, we can filter out the irrelevant path and infer the final answer.
>
> ## Weakness 3: Additional time cost for zero-shot generalizability.
>
> Zero-shot transfer experiments are conducted on three new datasets: FreebaseQA, CSQA and MedQA. GCR was applied directly to these new datasets without additional fine-tuning. It showed greater improvements on FreebaseQA and CSQA, underscoring its zero-shot generalizability. We hypothesize that the less significant gains observed on MedQA may stem from **LLMs having limited knowledge in the medical domain, which hampers their reasoning capabilities.**
>
> The additional time cost of implementing GCR mainly comes from the graph-constrained decoding. We present the additional time using different KG-specialized LLMs below. From the results, it is evident that the introduced additional time would decrease with the size of LLMs. Notably, the time can be further reduced with optimizations such as LLM quantization and flash attention.
>
> Additional time (s) introduced by GCR under different KG-specialized LLMs.
>
> | KG-specialized LLMs | Time (s) |
> | :---- | :---- |
> | Qwen2-0.5B | 1.8 |
> | Qwen2-1.5B | 2.3 |
> | Qwen2-7B | 4.4 |
> | Meta-3.1-8B | 3.6 |
>
> ## Weakness 4: Discuss and summarize more KG-enhanced methods
>
> Thanks for the suggestions. Due to the limited space, we add more discussion about existing KG-enhanced methods into Appendix A of the revision.

---

> > ### Author Response · Authors · 2024-11-18
> > **Official response by authors to Reviewer SPbz: Part 2**
> >
> > ## Question 1: Reasoning Across Multiple Paths
> >
> > Our GCR can well handle cases with multi-valid paths. As introduced in Section 4.4, GCR could take advantage of the GPU parallel computation to conduct multi-path explorations on KGs with beam-search. It could simultaneously generate $K$ reasoning paths and hypothesis answers with beam search in a single LLM call. The effectiveness of different $K$ is analyzed in Figure 4 where larger $K$ can lead to a better recall of the valid paths and answers. In addition, we compare the F1 performance under different numbers of ground-truth answers with RoG, which requires reasoning across multiple reasoning paths to find all answers. From the results, we can observe that GCR exhibits better performance in exploring multiple paths for reasoning.
> >
> > F1 comparison against RoG under different numbers of ground-truth answers.
> >
> > |  | WebQSP |  |  |  | CWQ |  |  |  |
> > | :---- | :---- | :---- | :---- | :---- | :---- | :---- | :---- | :---- |
> > | Methods | \# Ans \= 1 | 2 \<= \# Ans \<= 4 | 5 \<= \# Ans \<= 9 | \# Ans \>= 10 | \# Ans \= 1 | 2 \<= \# Ans \<= 4 | 5 \<= \# Ans \<= 9 | \# Ans \>= 10 |
> > | GCR | **71.31** | 78.14 | **83.47** | **63.20** | 55.80 | **64.08** | **62.57** | **55.32** |
> > | RoG | 67.89 | **79.39** | 75.04 | 58.33 | **56.9** | 53.73 | 58.36 | 43.62 |
> >
> > ## Question 2: Irrelevant Paths
> >
> > Thanks for bringing up this important question. Although GCR achieves 100% trustful reasoning, there are still some failure cases due to the noise and redundant information in KGs (also seen in our response to Q3 of reviewer rwhg). To alleviate the effects of unreliable paths, we propose the graph inductive reasoning module. We adopt a powerful general LLM to reason over multiple generated paths and select useful paths to produce final answers. Detailed examples can be found in our response to Weakness 2.
> >
> > ## Question 3: Analysis of Beam-size
> >
> > **A larger beam-size would lead to better recall of the valid reasoning paths and answers but slightly hampers the inference speed.** In GCR, we combine the advantage of the GPU parallel computation to conduct multi-path explorations on KGs with beam-search. It could simultaneously generate $K$ reasoning paths and hypothesis answers with beam search in a single LLM call. The effectiveness of different $K$ is analyzed in Figure 4 where larger $K$ can lead to a better recall of the valid paths and answers. Meanwhile, a larger beam size would also affect the inference speed, which can also be found in Figure 4\. However, such time can be further reduced with optimizations such as LLM quantization and flash attention.
> >
> > ## Question 4: Dynamic and Temporal KGs
> >
> > Thank you for highlighting this intriguing direction. Adapting the KG-Trie for temporal and dynamic knowledge graphs presents unique challenges. One potential solution is to incorporate time constraints when searching for paths. For instance, we can utilize temporal random walks \[1\] or temporal BFS \[2\] to extract paths from temporal KGs while preserving their temporal correlations. We can then encode these paths using the proposed KG-Trie to facilitate reasoning on temporal KGs. Exploring such a method to a dynamic and temporal KG is out of the current scope of this paper, and will be studied in future research.
> >
> > \[1\] Jin, M., Li, Y. F., & Pan, S. (2022). Neural temporal walks: Motif-aware representation learning on continuous-time dynamic graphs. Advances in Neural Information Processing Systems, 35, 19874-19886.
> > \[2\] Huang, S., Cheng, J., & Wu, H. (2014). Temporal graph traversals: Definitions, algorithms, and applications. arXiv preprint arXiv:1401.1919.

---

> ### Comment · Reviewer_SPbz · 2024-11-19
> **Official Response by Reviewer SPbz**
>
> I appreciate the authors' response to my concerns. However, after reading the response, I still have the following concerns:
> 1. **In practical applications, since users' inputs are unpredictable, I believe it is crucial to build a KG-Trie for all entities in the KG.** Therefore, considering the time and space costs of constructing such a KG-Trie is necessary. Additionally, given the unpredictability of the input, I remain curious: if the entities in the input question cannot be retrieved from the KG-Trie, does this mean that GCR cannot provide any gain in such scenarios?
>
> 2. In the general reply, the authors reported time results for L=1,2,3. However, I am curious about the implications of a larger L, such as L=10 for highly complex inference tasks. **Would the time cost grow exponentially with larger L, potentially limiting the practical applicability of GCR?**
>
> 3. Have the authors considered using larger KG-specialized LLMs, such as Llama 3 (13B or 70B)? How would the performance improvement compare with the associated increase in time costs?
>
> 4. I believe contamination in the KG is an important factor to consider, as even widely used sources like Wikipedia contain many incorrect links. **Therefore, I am somewhat concerned about the claim of "zero reasoning hallucination." I hope the authors can provide more detailed information on this aspect.**
>
> In conclusion, **I will maintain my current score** and look forward to the authors' further responses to these questions.

---

> > ### Author Response · Authors · 2024-11-20
> >
> > We sincerely thank the reviewer for their thoughtful comments and the opportunity for us to clarify the raised concerns! Below, we provide our responses to each question, hoping they could address your concerns. Please don't hesitate to reach out if you have any further questions.
> >
> > ## Question 1:  It is crucial to build a KG-Trie for all entities in the KG
> >
> > In industrial KG-enhanced QA applications, the unpredictability of user inputs is addressed by leveraging **entity recognition** **tools** such as OpenIE to identify question entities \[1\]. These entities are then linked to the corresponding entities in the KG with **entity-linking tools**. Based on these starting nodes, we could **retrieve the relevant subgraphs** and construct the KG-Trie in real-time. This approach ensures efficiency in both time and space while maintaining scalability.
> >
> > This real-time construction can be further optimized by caching frequently queried subgraphs to reduce repetitive computation. If the question entities cannot be found in the cached KG-Trie, we will conduct the aforementioned process to construct a corresponding KG-Trie to ensure the applicability of GCR. We have elaborated on this approach and provided examples in **Fig 6\. of Appendix B** of the revised manuscript.
> >
> > \[1\] Ant Group Knowledge Graph Team. KAG: Boosting LLMs in Professional Domains via Knowledge Augmented Generation. arXiv preprint arXiv:2409.13731.
> >
> > ## Question 2: Time cost implications for larger L (e.g., L=10)
> >
> > Thank you for raising this point. We acknowledge that directly constructing a KG-Trie for a larger $L$ can be time-consuming, especially for highly complex inference tasks. To address this, **GCR can be integrated with existing planning-based methods to decompose complex questions into multiple shorter steps** \[2\]. By breaking down the reasoning process, we can construct a KG-Trie with a smaller $L$ (e.g. 2 or 3) for each subtask to conduct reasoning, thereby reducing computational overhead while maintaining inference quality. This modular approach not only enhances scalability but also aligns with real-world applications where stepwise reasoning often mirrors human problem-solving.
> >
> > \[2\] Li, Y., et al. A Framework of Knowledge Graph-Enhanced Large Language Model Based on Question Decomposition and Atomic Retrieval. EMNLP 2024.
> >
> > ## Question 3: Using larger KG-specialized LLMs
> >
> > Thanks for your question. We have integrated larger KG-specialized LLMs, such as Llama-2-13B in experiments. Due to the limitation of GPUs, we cannot conduct training for 70B model right now. From the results, the performance increases as the model scales, which is consistent with the findings that larger LLMs exhibit stronger reasoning ability. However, the performance gain is less compared to the time growth. This indicates that it would be great to utilize lightweight KG-specialized LLMs in graph-constrained decoding while larger LLMs in graph-inductive reasoning to balance efficiency and effectiveness.
> >
> > Performance and time cost of different KG-specialized LLMs, ChatGPT is adopted as general LLM.
> > | KG-specialized LLMs | Time (s) | Hit |
> > | :---- | :---- | :---- |
> > | Qwen2-0.5B | 1.8 | 87.48 |
> > | Qwen2-1.5B | 2.3 | 89.21 |
> > | Qwen2-7B | 4.4 | 92.31 |
> > | Llama-2-7B | 3.9 | 92.55 |
> > | Llama-2-13B | 9.3 | 92.89 |
> >
> > ## Question 4:  Concerns about the claim of “zero reasoning hallucination”.
> >
> > Thanks for your suggestions. In this paper, we focus on the **KG-constrained Zero-hallucination** where the LLM-generated reasoning paths can be fully grounded within the KG. This ensures that the reasoning process aligns with real-world facts. Unlike noisy open-world knowledge, facts in KGs are usually verified, making them a reliable source for assessing the faithfulness of reasoning. This is consistent with prior work that uses KGs to assess the faithfulness of LLM reasoning \[2\]. Therefore, it is reasonable to classify reasoning as faithful or not based on the existence of paths in KGs. Under this definition, our experiments (Figure 5\) demonstrate that GCR fulfills the claim of "zero hallucination."
> >
> > However, we acknowledge that KGs are not free from incompleteness or incorrect facts, which can occasionally lead to false positives. Detecting such hallucinations without additional external evidence remains a challenge. To address this limitation, we plan to explore the integration of cross-references between multiple knowledge sources—such as KGs, web data, and documents—to further enhance the faithfulness of reasoning in future work. **To avoid over-claims, we have clarified the “KG-constrained Zero-hallucination” definition in Section 3 and its limitations in Appendix G of the revised paper.**
> >
> > \[3\] Thi Nguyen et al. 2024\. Direct Evaluation of Chain-of-Thought in Multi-hop Reasoning with Knowledge Graphs. ACL 2024.

---

> ### Comment · Reviewer_SPbz · 2024-11-24
>
> I appreciate the authors' response to my concerns. I still have some concerns regarding the construction of the kg-trie. If it does not require a complete kg-trie to be built, would user inputs in an open-ended setting risk not being captured or retrieved by the kg-trie even with the NER or EL tools? Given this, I remain focused on the issue of kg-trie construction time.  I am inclined to maintain the current scores for now. I am expecting that the authors to provide more details for me to reconsider my evaluation.

---

> > ### Author Response · Authors · 2024-11-25
> >
> > We sincerely thank the reviewer for their thoughtful comments and continued engagement with our discussion. We fully appreciate the concerns regarding the construction of the KG-Trie and the potential risks involved in user input understanding.
> >
> > Open-ended settings present additional challenges for retrieving information from knowledge graphs (KGs), but advanced **graph-retrieval techniques** can help mitigate these issues \[1\]. These techniques focus on extracting **small, question-related subgraphs** that contain the answer entities. Our KG-Trie can be efficiently built on these subgraphs and perform graph-constrained reasoning to derive the final answer.
> >
> > Among the graph-retrieval techniques, the NER (Named Entity Recognition) and EL (Entity Linking) techniques are widely adopted and well-studied in the industry due to their efficiency. To further capture the user’s intent, recent studies have utilized the pre-trained language model to retrieve facts from KGs without NER and EL \[2\]. Recent studies have also utilized GNN and LLM to find better subgraphs from KGs that capture user inputs \[3,4\].
> >
> > While graph-retrieval techniques may add some time, they significantly reduce graph size (e.g., fewer than 100 entities \[3\]), greatly enhancing KG-Trie construction efficiency. Our proposed methods can be integrated with any graph-retrieval techniques to facilitate efficient dynamic KG-Trie construction and support user input effectively. A detailed discussion on combining GCR with graph retrieval algorithms is provided in Appendix B.3.
> >
> > \[1\] Peng, B., et al (2024). Graph retrieval-augmented generation: A survey. arXiv preprint arXiv:2408.08921.
> > \[2\] Baek, J., et al. (2023). Direct fact retrieval from knowledge graphs without entity linking. ACL 2023\.
> > \[3\] He, X., et al, B. (2024). G-retriever: Retrieval-augmented generation for textual graph understanding and question answering. arXiv preprint arXiv:2402.07630.
> > \[4\] Luo, L., et al. (2023). Reasoning on graphs: Faithful and interpretable large language model reasoning. ICLR 2024\.

---

> > > ### Comment · Reviewer_SPbz · 2024-11-28
> > >
> > > I appreciate the author's response. However, I still believe that the open-end setting requires KG-trie time for any real-world application. The introduction of other methods in the author's reply does not alleviate my concerns. **Due to the lack of more detailed information on the construction time cost, I will maintain my current score.** I am considering that the full construction cost might be significantly higher than initially anticipated, and I believe this is an important issue for improvement.

---

> > > > ### Author Response · Authors · 2024-11-28
> > > >
> > > > Dear Reviewer SPbz,
> > > >
> > > > Thank you for your continued engagement with our work and for raising the important concern regarding the construction time of the KG-Trie in the open-ended setting. To address your concerns more effectively, **we have provided a detailed breakdown of the time consumption for each component involved in the KG-Trie construction**:
> > > >
> > > > | Component | Description | Implementation | Time (s) |
> > > > | :---- | :---- | :---- | :---- |
> > > > | Named Entity Recognition (NER) | Identify mentioned entities in user questions | [Spacy](https://spacy.io/api/entityrecognizer) | 0.0059 |
> > > > | Entity Linking (NL) | Link to entities in KGs | [ColBERTv2](https://huggingface.co/colbert-ir/colbertv2.0) | 0.0457 |
> > > > | Graph Retrieval | Retrieve question-relevant subgraphs for KG-Trie construction (Eq. 3). | 2-hop BFS implemented with SPARQL. | 0.0133 |
> > > > | Tokenizer | Tokenize paths into tokens for building LLM constraints (Eq. 4). | [Llama-3-8B Tokenizer implemented by Huggingface.](https://huggingface.co/meta-llama/Meta-Llama-3-8B-Instruct) | 0.1227 |
> > > > | Trie construction | Store the tokenized paths with Trie (Eq. 5). | [Python MARISA Trie](https://github.com/pytries/marisa-trie) | 0.0962 |
> > > > | **Total** |  |  | **0.2838** |
> > > >
> > > > As shown in the table, the overall time for constructing the KG-Trie under the open-end setting is approximately 0.28 seconds. This includes the time for all necessary stages, such as Named Entity Recognition, Entity Linking, graph retrieval, tokenization, and trie construction.
> > > >
> > > > We hope that this more detailed information helps alleviate your concerns. If you have any additional questions regarding the cost of any individual component, please do not hesitate to raise them. We understand that further investigation will be needed when implementing it in real-world applications, and we are committed to collaborating with our industrial partner to explore ways to optimize the time cost.
> > > >
> > > > We sincerely appreciate your thoughtful feedback and your continued engagement with our work.
> > > >
> > > > Best regards,
> > > > The Authors

---

### Official Review · Reviewer_hR19 · 2024-11-01

**Soundness:** 3
**Presentation:** 3
**Contribution:** 3
**Rating:** 6
**Confidence:** 4

**Summary:**

This paper introduces Graph-Constrained Reasoning (GCR) that stores KG paths in a Trie structure as constraints to guide the decoding process of LLMs and only generates reasoning paths that are valid in KGs. GCR combines a lightweight KG-specialized LLM for graph-constrained reasoning with a powerful general LLM for inductive reasoning, achieving good performance and zero-shot generalization on various KGQA benchmarks.

**Strengths:**

1. The motivation is clear and the proposed method is technically sound.

2. The manuscript is well-organized and easy to follow.

3. Extensive experiments have verified the effectiveness of the proposed method.

**Weaknesses:**

1. Preprocessing all paths appears to be both costly and potentially redundant. Could you discuss the space complexity involved in this process? Additionally, how do you plan to control the length of the reasoning paths to optimize efficiency?

2. Instead of using a Trie to constrain the LLM step by step, what are the implications of performing a post-validation with graph querying for the beam-search paths? How do these two approaches differ in terms of time and space complexity?

3. Given that the proposed method has achieved zero reasoning hallucination, analysing the error cases would provide deeper insights and make the results more convincing. Could you include such an analysis in the discussion?

**Questions:**

Please refer to Weaknesses.

---

> ### Author Response · Authors · 2024-11-18
> **Official response by authors to Reviewer hR19**
>
> We appreciate the reviewer’s positive comments. We have revised the manuscript based on your feedback and provided detailed responses to each point below. We hope our answers address your questions.
>
> ## Weakness 1: Computational Expense of KG-Trie Construction
>
> **We want to clarify that there is no need to build a KG-Trie for all entities in KGs.** In experiments, we only construct the KG-Trie for entities mentioned in questions. The KG-Trie can be either pre-computed or constructed on-demand to minimize pre-processing time. When the user’s questions are comping, we can identify the mentioned question entities and retrieve the question-related subgraphs from KGs for KG-Trie construction. This process is also very efficient, where the detailed analysis of time complexity and actual running time can be found in **our responses to all reviewers.** We also discuss the potential solutions to further improve the efficiency and scale into real-world applications with billion-scale KGs.
>
> The number of hops is determined by the size of the graphs and the distribution of questions. Empirically, larger hops can improve answer coverage but increase KG-Trie construction time as shown in our general responses. Therefore, users must balance efficiency and effectiveness. In our experiments, we set the hops to 2 or 3, as this covers 99% of question answers.
>
> ## Weakness 2: Difference with Post-validation of Paths
>
> We thank you for your inspiring comments. **While the post-validation can also verify the trustworthiness of the reasoning paths, it would bring significant additional computation costs and potential errors**, which might not be suitable for inference. Thi et al. \[1\] propose a post-validation strategy to verify the reasoning process by checking the existence of paths in KGs.
>
> However, due to the unstructured nature and randomness of the generated text, it is hard to match it with paths in KGs. Additional steps like entity recognition, relation extraction, and entity linking are required in the post-validation stage. Each of the steps requires different techniques and additional computation costs, resulting in higher latency. Moreover, the errors could propagate and undermine the trustworthiness. In contrast, our graph-constrained decoding ensures trustworthiness during decoding without additional validation costs.
>
> \[1\] Thi Nguyen, Linhao Luo, Fatemeh Shiri, Dinh Phung, Yuan-Fang Li, Thuy-Trang Vu, and Gholamreza Haffari. 2024\. Direct Evaluation of Chain-of-Thought in Multi-hop Reasoning with Knowledge Graphs. In Findings of the Association for Computational Linguistics: ACL 2024, pages 2862–2883, Bangkok, Thailand. Association for Computational Linguistics.
>
> ## Weakness 3: Analysis of Error Cases
>
> We appreciate your suggestion to include an analysis of error cases. Although our results show that GCR achieves 100% faithful reasoning, there are some failure cases where GCR generates incorrect answers.
>
> **Generated paths are unrelated to the questions:** Although LLMs exhibit strong reasoning ability, they still cannot always find meaningful paths to the answers. For example,
>
> > **Question:** what electorate does anna bligh representt?
> > **Ground-truth answer:** Electoral district of South Brisbane
> > **Generated paths:** Anna Bligh \-\> government.politician.government\_positions\_held \-\> m.0cr320w \-\> government.government\_position\_held.jurisdiction\_of\_office \-\> Queensland
> > **Predicted answer:** Queensland
>
> Although GCR provides a valid reasoning path that describes Anna Bligh's political position, it lacks information about her electoral district, resulting in incorrect answers.
>
> **KG incompleteness:** Although KGs store abundant factual knowledge, there are still missing facts. For example,
>
> > **Question:** who plays ken barlow in coronation street?
> > **Ground-truth answer:** William Roache
> > **Generated paths:** Coronation Street \-\> tv.tv\_program.program\_creator \-\> Tony Warren \-\> fictional\_universe.fictional\_character\_creator.fictional\_characters\_created \-\> Ken Barlow
> > **Predicted answer:**  Ken Barlow
>
> Because there is no information about the character's player stored in KGs, GCR cannot generate the correct answer. We will explore the reasoning for incomplete KGs in the future. These failure cases will be included in Appendix F.4 to discuss the limitations and potential future directions.

---

> > ### Comment · Reviewer_hR19 · 2024-11-23
> > **Response to authors**
> >
> > Thanks to the authors for their detailed responses. I have no further questions.

---

> > > ### Author Response · Authors · 2024-11-24
> > >
> > > Dear Reviewer hR19,
> > >
> > > Thank you very much for the quick and prompt response! We are happy to know that your concerns have been addressed.
> > >
> > > Meanwhile, we would greatly appreciate it if you could consider upgrading your rating to acknowledge our responses. Your invaluable suggestions greatly enhance the quality of the paper.
> > >
> > > Sincerely,
> > >
> > > The Authors

---

### Official Review · Reviewer_rwhg · 2024-11-03

**Soundness:** 2
**Presentation:** 2
**Contribution:** 2
**Rating:** 3
**Confidence:** 4

**Summary:**

The paper introduces graph-constrained reasoning (GCR), a novel framework that bridges structured knowledge in KGs with unstructured reasoning in LLMs. To eliminate hallucinations, GCR ensures faithful KG-grounded reasoning by integrating KG structure into the LLM decoding process through KG-Trie, a trie-based index that encodes KG reasoning paths.

**Strengths:**

The approach is interesting.
The experiments show strong results. The paper is well written though the organization of the approach description can be improved.

**Weaknesses:**

1. The authors should give some cases in which GCR has a Faithful Reasoning Path and RoG does not. The key reason that GCR outperforms RoG was not fully explained. RoG seems a simplified version of GCR.

2. The tokenizer-level decoding method may lead to entities or relationships that cannot be recognized in KGs, which in turn invalidates the model. Meanwhile, the tokenizer-level decoding will remarkably
 increase the runtime compared to the methods of the same type (see Table2).

3. Is the definition of faithful reasoning in section 5.2 sound? This definition is a bit broad since some wrong paths can also lead to right answers as many papers point out.


4. The construction of KG-trie is still time-consuming and takes up space if you want to cover all the questions.

**Questions:**

1. The motivation for using BFS and Tokenizer in E.4 is not clear. Can use DFS and word-level Tokenizer work?

2. The authors used simple scenarios in their experiments, but is the model as efficient as it claims when there are many candidate paths and it takes multiple hops to reach the target entity?

3. Can the author analyze the reasons for unfaithful reasoning in GCR and give some examples?

---

> ### Author Response · Authors · 2024-11-18
> **Official response by authors to Reviewer rwhg: Part 1**
>
> Thank you for your detailed review and constructive feedback. We appreciate your insightful comments, which have provided valuable guidance for improving our work. Below, we address your main concerns by clarifying some misunderstandings.
>
> ## Weakness 1: Clarification on GCR’s Advantage over RoG
>
> We want to clarify the advantages of GCR over RoG in the following aspects:
>
> **Integration of KG-Trie**: RoG uses a planning-retrieval framework, where reasoning paths are retrieved from knowledge graphs (KGs) based on plans generated by large language models (LLMs). However, the lack of constraints in LLMs can lead to hallucinations, resulting in 33% invalid plans, as shown in Fig. 1\. In contrast, GCR integrates a KG-Trie into the LLM decoding process without retrieval, ensuring that only valid KG-based paths are produced. This method prevents hallucinations and maintains high reasoning accuracy, demonstrated by the 100% faithful reasoning ratio in Figure 5\.
>
> **Combination of LLMs**: GCR leverages both a lightweight KG-specialized LLM and a powerful general LLM, combining their strengths for constrained graph reasoning and inductive reasoning. This dual approach enables GCR to explore multiple reasoning paths efficiently and provide more accurate answers.
>
> ## Weakness 2: Tokenizer-Level Decoding Concerns
>
> **We want to clarify that our token-level graph-constrained decoding would not lead to entities or relationships that do not exist in KGs.** During decoding, we use the KG-Trie to restrict the tokens generated by the LLM to those starting with valid prefixes stored in the Trie. This approach has been used by previous methods to limit LLM output within a specific scope, such as all entities in KGs \[1\]. Our KG-Trie is constructed from paths within KGs. Therefore, under these constraints, only valid entities and relations from KGs can be generated by LLMs to form reasoning paths. We have thoroughly checked the generated results and found **zero invalid entities or relations**, as shown in Figure 5\.
>
> Meanwhile, **the token-level graph-constrained decoding is more efficient and effective than other LLM-based graph reasoning methods.** Due to the unstructured nature of LLMs, they are difficult to apply directly for reasoning on structured knowledge graphs (KGs). Previous LLM-based graph reasoning methods, such as ToG \[2\], typically follow an agent paradigm where LLMs iteratively query information from KGs. This approach incurs multiple API calls, resulting in high computational costs and latency. With KG-Trie, we enable LLMs to reason on KGs within a single decoding process, significantly reducing computation overhead and latency. Additionally, incorporating KG-Trie into LLM decoding does not introduce extra computational costs since it only masks out the probabilities of invalid tokens. Furthermore, this integration leverages GPU parallel computation to traverse multiple paths using beam search.
>
> Table 2 shows that GCR requires less running time and fewer LLM calls than LLM agent-based methods, such as ToG. While retriever-based methods are slightly more efficient than GCR, their performance is limited by the accuracy of additional retrievers, leading to worse results compared to GCR.
>
> Efficiency Comparison with Agent-based LLM methods.
>
> | Methods | Avg. Runtime (s) | Avg. \#LLM Calls | Avg. \# LLM Tokens |
> | :---- | :---- | :---- | :---- |
> | ToG (ChatGPT) | 16.14 | 11.6 | 7,069 |
> | GCR | 3.60 | 2 | 231 |
>
> \[1\] Nicola De Cao, Gautier Izacard, Sebastian Riedel, and Fabio Petroni. Autoregressive entity retrieval.
> In International Conference on Learning Representations, 2022\.
> \[2\] Sun, J., Xu, C., Tang, L., Wang, S., Lin, C., Gong, Y., ... & Guo, J. Think-on-Graph: Deep and Responsible Reasoning of Large Language Model on Knowledge Graph. In The Twelfth International Conference on Learning Representations.
>
> ## Weakness 3: Definition of Faithful Reasoning
>
> Thank you for the inspiring comments. In this paper, we define faithful reasoning as generating paths that can be found within KGs, ensuring that the reasoning process aligns with real-world facts. Unlike noisy open-world knowledge, KGs contain abundant factual information verified by experts, which has been used to assess the faithfulness of LLM reasoning \[3\]. Therefore, it is reasonable to classify reasoning as faithful or not based on the existence of paths in KGs. While KGs are incomplete and some valid paths may not be present (false negatives), we will explore this further in future work.
>
> \[3\] Thi Nguyen, Linhao Luo, Fatemeh Shiri, Dinh Phung, Yuan-Fang Li, Thuy-Trang Vu, and Gholamreza Haffari. 2024\. Direct Evaluation of Chain-of-Thought in Multi-hop Reasoning with Knowledge Graphs. In Findings of the Association for Computational Linguistics: ACL 2024, pages 2862–2883, Bangkok, Thailand. Association for Computational Linguistics.

---

> > ### Author Response · Authors · 2024-11-18
> > **Official response by authors to Reviewer rwhg: Part 2**
> >
> > ## Weakness 4: Computational Expense of KG-Trie Construction
> >
> > **We want to clarify that there is no need to build a KG-Trie for all entities in KGs.** In experiments, we only construct the KG-Trie for entities mentioned in questions. The KG-Trie can be either pre-computed or constructed on-demand to minimize pre-processing time. When the user’s questions are coming, we can identify the mentioned question entities and retrieve the question-related subgraphs from KGs for KG-Trie construction. This process is also very efficient, where the detailed analysis of time complexity and actual running time can be found in **our responses to all reviewers.** We also discuss the potential solutions to further improve the efficiency and scale into real-world applications with billion-scale KGs.
> >
> > ## Question 1: Can we use DFS and word-level Tokenizer?
> >
> > Yes, we can use DFS in KG-Trie construction since it explores paths up to a maximum length of $L$ starting from specific entities, sharing the same complexity as BFS. We also discuss the potential of using other efficient graph traversal algorithms, such as random walk for KG-Trie construction, which is detailed in our responses to all reviewers.
> >
> > However, we cannot simply use a word-level tokenizer because GCR aims to conduct KG reasoning via LLM decoding reasoning paths which are generated tokens by tokens. Therefore, we adopt the same token-level tokenizers used in LLMs. A word-level tokenizer can only be used if desired by the LLMs. More detailed motivations for KG-Trie are provided in our responses to all reviewers.
> >
> > ## Question 2: Multi-path and Multi-hop Reasoning.
> >
> > **Multi-path Explorations:** As noted in Section 4.4, GCR leverages GPU parallelism for multi-path KG exploration with beam-search. Figure 4 in the paper shows that higher $K$ improves answer recall. Besides, we compare with RoG under different numbers of ground-truth answers, which requires reasoning across multiple reasoning paths. Compared to RoG, GCR achieves better F1 performance by effectively reasoning over multiple paths.
> >
> > F1 comparison against RoG under different numbers of ground-truth answers.
> > |  | WebQSP |  |  |  | CWQ |  |  |  |
> > | :---- | :---- | :---- | :---- | :---- | :---- | :---- | :---- | :---- |
> > | Methods | \# Ans \= 1 | 2 \<= \# Ans \<= 4 | 5 \<= \# Ans \<= 9 | \# Ans \>= 10 | \# Ans \= 1 | 2 \<= \# Ans \<= 4 | 5 \<= \# Ans \<= 9 | \# Ans \>= 10 |
> > | GCR | **71.31** | **78.14** | **83.47** | **63.20** | 55.80 | **64.08** | **62.57** | **55.32** |
> > | RoG | 67.89 | **79.39** | 75.04 | 58.33 | **56.9** | 53.73 | 58.36 | 43.62 |
> >
> > **Multi-hop Reasonings:** To demonstrate the effectiveness of multi-hop reasonings. We illustrate the F1 performance under different hops. From results, we can observe that GCR also outperforms baselines in multi-hop reasoning.
> >
> > F1 comparison against RoG under different hops of reasoning.
> > |  | WebQSP |  |  | CWQ |  |  |
> > | :---- | :---- | :---- | :---- | :---- | :---- | :---- |
> > | Methods | 1 hop  | 2 hop | \>=3 hop | 1 hop  | 2 hop | \>=3 hop |
> > | GCR | 75.05 | **72.72** | \- | **64.54** | **62.44** | **43.82** |
> > | RoG | **77.03** | 64.86 | \- | 62.88 | 58.46 | 37.82 |
> >
> > These additional results have been added to Appendix F.2 and F.3 of the revision.
> >
> > ## Question 3: Analysis and Examples of Unfaithful Reasoning in GCR.
> >
> > We want to clarify that **there is no unfaithful reasoning in GCR** under the definition of faithful reasoning in Section 5.2. Because all the generated reasoning paths are grounded in KGs. However, there are some failure cases where GCR generates incorrect answers.
> >
> > **Generated paths are unrelated to the questions:** Although LLMs exhibit strong reasoning ability, they still cannot always find meaningful paths to the answers. For example,
> >
> > > Question: what electorate does anna bligh representt?
> > > Ground-truth answer: Electoral district of South Brisbane
> > > Generated paths: Anna Bligh \-\> government.politician.government\_positions\_held \-\> m.0cr320w \-\> government.government\_position\_held.jurisdiction\_of\_office \-\> Queensland
> > > Predicted answer: Queensland
> >
> > Although GCR provides a valid reasoning path that describes Anna Bligh's political position, it lacks information about her electoral district, resulting in incorrect answers.
> >
> > **KG incompleteness:** The knowledge graphs are incomplete with some missing facts.
> >
> > > Question: who plays ken barlow in coronation street?
> > > Ground-truth answer: William Roache
> > > Generated paths: Coronation Street \-\> tv.tv\_program.program\_creator \-\> Tony Warren \-\> fictional\_universe.fictional\_character\_creator.fictional\_characters\_created \-\> Ken Barlow
> > > Predicted answer:  Ken Barlow
> >
> > Because there is no information about the character's player stored in KGs, GCR cannot generate the correct answer. We will explore the reasoning for incomplete KGs in the future. These failure cases will be included in Appendix F.4 to discuss the limitations and potential future directions.

---

### Official Review · Reviewer_tHQb · 2024-11-04

**Soundness:** 2
**Presentation:** 3
**Contribution:** 2
**Rating:** 5
**Confidence:** 4

**Summary:**

This paper proposes graph-constrained reasoning (GCR). GCR integrates KG structure into the LLM decoding process through KG-Trie, a trie-based index that encodes KG reasoning paths. It leverages a lightweight KG specialized LLM for graph constrained reasoning alongside a powerful general LLM for inductive reasoning over multiple reasoning paths.

**Strengths:**

1, The proposed method achieves the 100% faithful reasoning. All the supporting reasoning paths can be found on KGs as shown in Figure 5.

2, The proposed method requires less average LLM tokens and small numbers of LLM calls during inference as shown in Table 2.

**Weaknesses:**

1, The construction of KG-Trie may be computationally expensive in the pre-processing. Although the inference of the proposed method is efficient, the overhead of the preprocessing seems time-consuming. The BFS in formula 3 extract all
-length edges around the
 in the preprocessing stage. In a large graph with millions of entities and edges, this will be expensive. These steps (formula 3-5) need to be done for every entity since we do not know the query entity in advance. In this paper authors choose
. In multi-hop reasoning on KGs a larger
 is needed which brings more preprocessing overhead. Can the authors explain the time complexity of preprocessing with both theoretical analysis and empirical results?

2, Missing graph reasoning baselines. In the experiments in Table 1 the graph reasoning baselines are included. However, some SOTA link prediction GNN methods like NBFNet[1] and ULTRA[2] are not in the table. These methods can be applied on Freebase and ConceptNet and should be included.

3, Experiments on more datasets are needed to show the superiority of the proposed method. In Table 1 only two datasets WebQSP and CWQ are included. On CWQ, GCR performs well. However on WebQSP, GCR only shows a small margin over GNN-RAG + RA. In Table 6 only two datasets CSQA and MedQA are includes. And the improvements on MedQA is not signifcant. More experiments on more datasets will make the proposed method more convincing.

[1] Zhu, Zhaocheng, et al. "Neural bellman-ford networks: A general graph neural network framework for link prediction." Advances in Neural Information Processing Systems 34 (2021): 29476-29490.

[2] Galkin, Mikhail, et al. "Towards foundation models for knowledge graph reasoning." arXiv preprint arXiv:2310.04562 (2023).

**Questions:**

Please refer to the questions mentioned in the Weaknesses part.

---

> ### Author Response · Authors · 2024-11-17
> **Official response by authors to Reviewer tHQb**
>
> We sincerely appreciate your thorough review and valuable feedback on our submission. We have provided a detailed response to each comment below. We hope our answers can properly address your concerns.
>
> ## Weakness 1: Computational Expense of KG-Trie Construction
>
> **We want to clarify that there is no need to build a KG-Trie for all entities in KGs.** In experiments, we only construct the KG-Trie for entities mentioned in questions. The KG-Trie can be either pre-computed or constructed on-demand to minimize pre-processing time. When the user’s questions are coming, we can identify the mentioned question entities and retrieve the question-related subgraphs from KGs for KG-Trie construction. This process is also very efficient, where the detailed analysis of time complexity and actual running time can be found in **our responses to all reviewers.** We also discuss the potential solutions to further improve the efficiency and scale into real-world applications with billion-scale KGs.
>
> ## Weakness 2: Inclusion of Graph Reasoning Baselines
>
> We appreciate your suggestion to include additional state-of-the-art graph reasoning methods. However, we want to mention that some GNN reasoning models, like NBFNet and ULTRA, cannot be easily adapted to the question-answering task. NBFNet and ULTRA are designed for inductive knowledge graph completion tasks, which cannot handle the richer semantics in the user’s natural language questions to predict the possible answers.
>
> To compare with GNN-based methods, we select several baselines that utilize the power of GNN in question answering, which are illustrated below. From the results, it is evident that our GCR outperforms all the baselines, demonstrating the superiority of LLMs in graph reasoning. These baselines are included in the graph reasoning section of Table 1\.
>
> Comparison with GNN-based graph reasoning baselines.
>
> |  | WebQSP |  | CWQ |  |
> | :---- | :---- | :---- | :---- | :---- |
> | Methods | Hit | F1 | Hit | F1 |
> | GraftNet \[1\] | 66.7 | 62.4 | 36.8 | 32.7 |
> | UniKGQA \[2\] | 77.2 | 72.2 | 51.2 | 49.1 |
> | ReaRev \[3\] | 76.4 | 70.9 | 52.9 | 47.8 |
> | GCR (Llama-3.1-8B \+ ChatGPT)  | **92.6** | 73.2 | 72.7 | 60.9 |
> | GCR (Llama-3.1-8B \+ GPT-4o-mini) | 92.2 | **74.1** | **75.8** | **61.7** |
>
> \[1\] Sun, H., Dhingra, B., Zaheer, M., Mazaitis, K., Salakhutdinov, R., & Cohen, W. (2018). Open Domain Question Answering Using Early Fusion of Knowledge Bases and Text. In Proceedings of the 2018 Conference on Empirical Methods in Natural Language Processing (pp. 4231-4242).
> \[2\] Jiang, J., Zhou, K., Zhao, X., & Wen, J. R. UniKGQA: Unified Retrieval and Reasoning for Solving Multi-hop Question Answering Over Knowledge Graph. In The Eleventh International Conference on Learning Representations.
> \[3\] Mavromatis, C., & Karypis, G. (2022, December). ReaRev: Adaptive Reasoning for Question Answering over Knowledge Graphs. In Findings of the Association for Computational Linguistics: EMNLP 2022 (pp. 2447-2458).
>
> ## Weakness3:  Limited Datasets
> Your feedback on the number of evaluated datasets is appreciated. To further demonstrate our method’s efficacy across a broader range of benchmarks and strengthen its contributions, we extend our zero-shot experiments to **additional datasets:** **FreebaseQA** \[4\]. The results are presented below. The FreebaseQA is another question-answering dataset based on Freebase knowledge graphs. From the results, we can observe that GCR achieves significant improvements in FreebaseQA, demonstrating its generalizability and transferability.
>
> Zero-shot transferability to other KGQA datasets.
>
> | Model | CSQA | MedQA | FreebaseQA |
> | :---- | :---- | :---- | :---- |
> | ChatGPT | 79 | 64 | 85 |
> | GCR (ChatGPT) | **85** | **66** | **93** |
> | GPT-4o-mini | 91 | 75 | 89 |
> | GCR (GPT-4o-mini) | **95** | **79** | **94** |
>
> \[4\] K. Jiang, D. Wu and H. Jiang, "FreebaseQA: A New Factoid QA Data Set Matching Trivia-Style Question-Answer Pairs with Freebase," Proc. of North American Chapter of the Association for Computational Linguistics (NAACL), June 2019.

---

### Author Response · Authors · 2024-11-17
**General reply to all reviewers about the efficiency of KG-Trie construction: Part 1**

We sincerely appreciate your thorough reviews and valuable feedback on our submission. We have noted your primary concerns regarding the efficiency and preprocessing overhead of constructing the KG-Trie. Below, we provide a comprehensive response to address these points and clarify our approach. Specifically, we first analyze the time and space complexity of KG-Trie construction. Then, we introduce several strategies to further improve efficiency and support real-world billion-scale KGs.

## Motivations of KG-Trie construction

Large language models (LLMs) have demonstrated remarkable reasoning capabilities through token-by-token decoding. However, the unstructured nature of LLMs poses challenges for conducting efficient reasoning over structured knowledge graphs (KGs). The KG-Trie addresses this challenge by converting KG structures into the format that LLMs can handle. It has been incorporated into the LLM decoding process as constraints, allowing for faithful reasoning paths that align with the graph’s structure.

## KG-Trie Construction Strategies

**We want to clarify that there is no need to build a KG-Trie for all entities in KGs.** The KG-Trie can be either pre-computed for fast inference or constructed on-demand to minimize pre-processing time. Users can choose to build the KG-Trie offline, allowing them to be used during inference at no additional cost. Alternatively, **we can only retrieve the question-related subgraphs around the question entities and construct a question-specific KG-Trie on-demand.** In experiments, we only construct the KG-Trie for entities mentioned in questions. Users can also develop their own strategies (e.g. dynamic cache) to balance pre-processing and inference overhead.  We have clarified our discussion about KG-Trie construction in Section 4.2 and Section 5.1 of the revision, and present a framework of cache-based KG-Trie construction in Appendix B.

## Time and Space Complexity of KG-Trie Construction

As discussed, it is not necessary to construct KG-Trie for all entities in KGs. Thus, we want to highlight that the time and space complexity for KG-Trie construction is affordable and can be easily improved in industry-level applications to support billions of scale graphs. To support this, we provide detailed theoretical analysis and empirical evidence.

**Theoretical Analysis**

* **Time Complexity**: Constructing the KG-Trie involves a BFS traversal to explore paths up to a maximum length of $L$ starting from certain entities. The time complexity of this traversal is $O(E^{L})$, where $E$ is the average number of edges per entity, and $L$ is the maximum path length. BFS ensures that all reachable paths up to length $L$ are considered. However, BFS can be replaced with other efficient graph-traversing algorithms, such as random walk \[1\] to further improve efficiency.
* **Space Complexity:** The space complexity of the KG-Trie depends on the number of unique paths and their tokenized representations. In the worst case, the space complexity is $O(E^L \\times T)$, where $T$ represents the average number of tokens per path. Trie structures are efficient for storing shared prefixes, which reduces redundancy and optimizes memory usage. Moreover, it supports efficient traversal of reasoning paths in constant time.

**Empirical Analysis**

We have provided the average BFS running time and space consumption of the KG-Trie construction to demonstrate its efficiency.

System settings:

- CPU: Intel(R) Xeon(R) Silver 4214R CPU @ 2.40GHz.
- Memory: 32G.
- BFS implementation: Virtuoso SPARQL
- Space storage: Pickle

In the experiment, we build the KG-Trie for all question entities of WebQSP dataset and measure the average running time and space consumption. The BFS is executed on the Freebase KG stored in a Virtuoso database. We retrieve the $L$-hop paths, then save the constructed KG-Trie with Pickle. The statistics show that both running time and space usage are acceptable, which highlights efficiency in KG-Trie construction. The results are also presented in Table 7 of Appendix B.2.2 in the revision.

Although a larger hop can lead to better coverage of the possible answer, it would significantly increase the time and space complexity. Thus, we set hops to 2 or 3 in experiments to balance between efficiency and effectiveness. Notably, time can be further reduced by utilizing multi-threading. Space consumption can be optimized by storing data in a database.

Average running time and space utilization of the KG-Trie construction.

| Hop | Avg. Running Time (s) | Space (Mb) |
| :---- | :---- | :---- |
| L=1 | 0.0058 | 0.4 |
| L=2 | 0.0133 | 0.5 |
| L=3 | 0.0219 | 2.5 |

\[1\] Xia, Feng, et al. "Random walks: A review of algorithms and applications." IEEE Transactions on Emerging Topics in Computational Intelligence 4.2 (2019): 95-107.

---

> ### Author Response · Authors · 2024-11-17
> **General reply to all reviewers about the efficiency of KG-Trie construction: Part 2**
>
> As the KG-Trie is independently constructed for each entity, it can be easily scaled with parallel processing. We provide the total running time of constructing 2-hop KG-Trie of all question entities in WebQSP dataset to show the improvement of parallel processing. It shows that the efficiency can be greatly improved with parallel processing. This parallel nature enables it to be executed on distributed computing systems such as Hadoop and Spark in real-world applications. More strategies to further improve efficiency and support real-world billion-scale KGs can be found in Appendix B.3 and B.4 of the revision.
>
> Total running time and improvement under different processing threads.
>
> | Total time (s) | Total Time (min) | Improvement |
> | :---- | :---- | :---- |
> | Thread=1 | 4.03 | 100% |
> | Thread=4 | 3.21 | 126% |
> | Thread=10 | 2.31 | 174% |
> | Thread=20 | 1.92 | 210% |

---

### Author Response · Authors · 2024-11-18
**General response to all reviewers**

We appreciate the detailed feedback from all reviewers. We have revised the paper accordingly, with edits highlighted in **BLUE**, and included detailed responses below. Here, we summarize the major revisions and address each comment. We hope our responses address your concerns.

1. To address the concerns of all reviewers regarding KG-Trie construction efficiency, we have carefully clarified our discussion in Sections 4.2 and 5.1 of the revision. We also presented a detailed complexity analysis, empirical studies, and a framework of cache-based KG-Trie construction in Appendix B.
2. To address reviewer tHQb's comments, we have revised the experiment results in Table 1 by adding more GNN-based graph reasoning baselines.
3. To address reviewer tHQb's comments, we have revised the zero-shot experiments in Table 6 of Section 5.4 by extending them to additional datasets: FreebaseQA.
4. To address the comments of reviewers rwhg and SPbz, we have added analyses about the performance of GCR under multi-path explorations and multi-hop reasoning in Appendix F.2 and F.3, respectively.
5. To address the comments of reviewers rwhg, hR19, and SPbz, we have added analyses about the failure cases predicted by GCR in Appendix F.4 to further discuss the limitations and future directions.

---

### Meta-Review · Area_Chair_Voec · 2024-12-17

**Metareview:**

Graph-constrained reasoning (GCR) is introduced to integrate knowledge graph (KG) structure into the LLM decoding process. It leverages a KG-specialized LLM for graph-constrained reasoning and a general LLM for inductive reasoning over multiple reasoning paths. To alleviate hallucinations, GCR utilizes KG-grounded reasoning by imposing KG structure into the LLM decoding process through KG-Trie, a trie-based index encoding KG reasoning paths.

While some reviewers acknowledged that the paper is well-written and the approach is reasonable, there are shared concerns about the paper:\
(1) The KG-trie construction process is inefficient and time-consuming.\
(2) The proposed method's advantages are not significantly distinct from other retrieval-based methods.\
(3) It is hard to ensure that all generated reasoning paths are strongly relevant to the question. There can be a risk of introducing irrelevant information.\
(4) Given that real-world KGs are often incomplete and contaminated, a more elaborated method should be presented to show how the proposed method works when entities are absent from the KG or when the reasoning paths generated contain unreliable logical pathways.

Additional Comments:\
The authors claimed, "To eliminate hallucinations, GCR ensures faithful KG-grounded reasoning." On the other hand, the authors also argue that "Thus, based on the internal knowledge of powerful LLMs, we can filter out the irrelevant path." This sounds like the authors are only taking upsides of KGs and LLMs, which is too optimistic; the authors say KGs are used to eliminate LLMs' hallucinations, while LLMs filter out irrelevant paths on KGs. What happens if KG's irrelevant paths fail to eliminate hallucinations and LLMs' hallucinations fail to filter out irrelevant paths on KGs?

**Additional Comments On Reviewer Discussion:**

All reviewers raised valid points, and there is some consensus about the paper's main weaknesses, summarized in the metareview. Reviewers tHQb and SPbz, in particular, provided detailed reviews and asked the authors additional questions. Although the authors answered the reviewers' points, it seems necessary for them to consider making significant revisions to their proposed method to make it more complete.

---

### Decision · Program_Chairs · 2025-01-22

Reject